# OPTIMA: OPTIMIZING EFFECTIVENESS AND EFFICIENCY FOR LLM-BASED MULTI-AGENT SYSTEM

## ABSTRACT

Large Language Model (LLM) based multi-agent systems (MAS) show remarkable potential in collaborative problem-solving, yet they still face critical challenges: low communication efficiency, poor scalability, and a lack of effective parameter-updating optimization methods for multi-agent collaboration. We present **OPTIMA**, a novel framework that addresses these issues by significantly enhancing *both* communication efficiency and task effectiveness in LLM-based MAS through LLM training. At its core, OPTIMA employs an *iterative generate, rank, select, and train* paradigm, incorporating a reward function that balances task performance, token efficiency, and communication readability. We explore various RL algorithms, including Supervised Fine-Tuning, Direct Preference Optimization, and their hybrid approaches, providing insights into their effectiveness-efficiency trade-offs for iterative LLM-based MAS training. Additionally, we integrate Monte Carlo Tree Search-inspired techniques for DPO data generation, conceptualizing conversation turns as tree nodes to explore diverse interaction trajectories. We evaluate OPTIMA on common multi-agent tasks, including information-asymmetric question answering and complex reasoning. Our method demonstrates consistent and substantial improvements over single-agent baselines and vanilla MAS based on Llama 3 8B, achieving up to *2.8x performance gain with less than 10% tokens* on tasks requiring heavy multi-agent information exchange. Moreover, OPTIMA's efficiency gains open new possibilities for leveraging inference-compute more effectively, potentially leading to improved inference-time scaling laws. By addressing fundamental challenges in multi-agent collaboration and providing a novel optimization framework, OPTIMA shows the potential towards scalable, efficient, and effective LLM-based MAS.

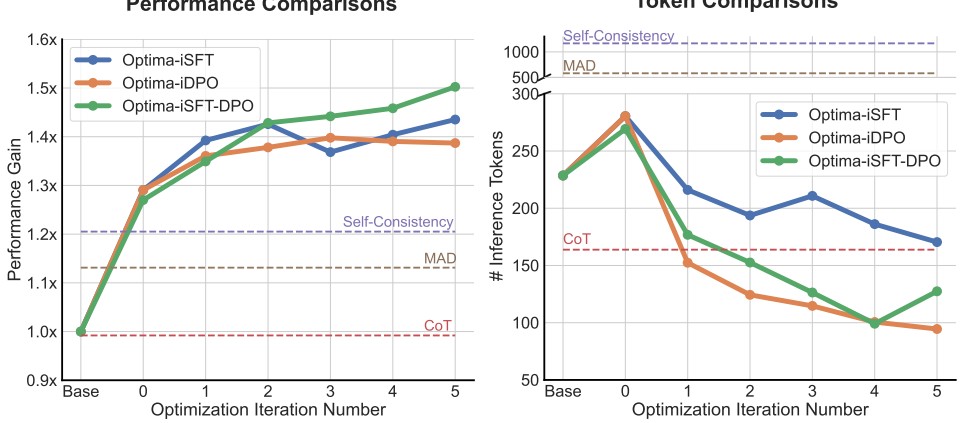

Figure 1: **Performance and efficiency of OPTIMA variants across optimization iterations. Left**: Average performance gain over iterations. OPTIMA variants consistently outperform CoT, Multi-Agent Debate (MAD), and Self-Consistency. **Right**: Average inference token numbers over iterations. All OPTIMA variants achieve better performance with substantially fewer tokens.

## 1 INTRODUCTION

Large Language Models (LLMs) have emerged as powerful tools for a wide range of tasks, from natural language processing to complex reasoning (OpenAI, 2023; Reid et al., 2024; Anthropic, 2024). A promising direction in leveraging these models is the development of autonomous multi-agent systems (MAS), which aim to harness the collective intelligence of multiple LLM-based agents for collaborative problem-solving and decision-making (Liang et al., 2023; Wang et al., 2024b; Du et al., 2024; Zhuge et al., 2024). However, for LLM-based MAS to be truly effective, they must overcome two critical challenges: **(a)** achieving efficient inter-agent communication to minimize computational costs, and **(b)** optimizing the collective performance of the system as a cohesive unit.

Current LLM-based MAS face significant difficulties in meeting these challenges. The coordination and communication between agents often lack efficiency, resulting in verbose exchanges that lead to increased token usage, longer inference times, and higher computational costs (Li et al., 2024b). This inefficiency is exacerbated by the *length bias* inherent in LLMs due to alignment training (Saito et al., 2023; Dubois et al., 2024), which favors longer responses even when concise communication would suffice (Chen et al., 2024d). Moreover, while recent work has explored training LLMs for single-agent tasks (Song et al., 2024; Xiong et al., 2024) and MAS training is well-studied in reinforcement learning (Johnson et al., 2000; Lanctot et al., 2017; Baker et al., 2020), there remains a lack of parameter-updating methods specifically designed to optimize LLM-based MAS as a unified system. Existing approaches primarily rely on simple agent profile evolution (Chen et al., 2024b) or memory evolution (Qian et al., 2024a;b; Gao et al., 2024), which fail to address the core issues of communication efficiency and collective optimization.

**Can we develop a training framework that simultaneously enhances the communication efficiency and task effectiveness of LLM-based MAS?** To address this question, we introduce **OP-TIMA**, an effective framework designed to optimize LLM-based MAS. At the heart of OPTIMA is an iterative *generate, rank, select, and train* paradigm, incorporating a reward function that balances task performance, token efficiency, and communication interpretability. This approach enables the development of MAS that are not only effective and efficient but also maintain interpretable communication patterns. Based on the reward function, OPTIMA leverages a combination of techniques to induce efficient and effective communication behaviors in LLM-based agents, including Supervised Fine-Tuning (SFT) (Zelikman et al., 2022; Gülçehre et al., 2023; Aksitov et al., 2023) and Direct Preference Optimization (DPO) (Rafailov et al., 2023; Pang et al., 2024), along with their hybrid variants. Furthermore, OPTIMA introduces an integration of Monte Carlo Tree Search (MCTS)-inspired techniques for DPO data generation, conceptualizing conversation turns as tree nodes to explore diverse interaction trajectories efficiently.

Importantly, by substantially reducing the number of tokens required for inference, OPTIMA not only improves computational efficiency but also opens new possibilities for leveraging inference-compute more effectively. This reduction in token usage allows for more samples within the same computational constraints, potentially leading to *better inference-time scaling laws*. As recent work has shown the importance of inference-time compute in improving model performance (Wu et al., 2024; Brown et al., 2024; Chen et al., 2024a), OPTIMA's efficiency gains could be combined with techniques like majority voting (Wang et al., 2023), leading to more effective LLM systems.

We evaluate OPTIMA on a diverse set of tasks spanning two multi-agent settings: **(a)** information exchange, including information-asymmetric question answering (Chen et al., 2024d; Liu et al., 2024), and **(b)** debate, encompassing mathematical and reasoning tasks (Du et al., 2024; Chen et al., 2024b; Wu et al., 2023). Using Llama 3 8B (Meta, 2024) as our base model, we demonstrate that OPTIMA consistently outperforms both single-agent MAS baselines, achieving up to 90% reduction in token usage and 2.8x increase in task performance.

To summarize, our main contribution is OPTIMA, a novel training framework that simultaneously optimizes *communication efficiency* and *task effectiveness*. To enhance high-quality training data generation *in multi-agent settings* for DPO, we introduce an integration of MCTS-like techniques. Our comprehensive empirical evaluation across diverse tasks demonstrates notable advancements in *both* token efficiency and task performance, while also providing insights into the learned communication patterns. Additionally, we examine the implications of OPTIMA's efficiency gains for inference-time scaling laws, underscoring its potential to improve the overall capabilities of LLM systems by enabling more effective utilization of inference-compute. By addressing the dual chal-

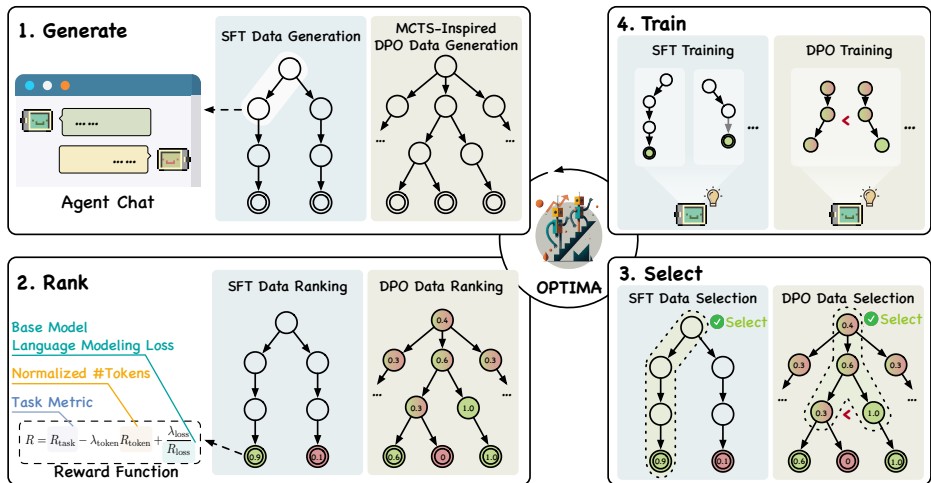

Figure 2: **Overview of the OPTIMA framework for training LLM-based MAS**. The iterative process includes four stages: *Generate, Rank, Select*, and *Train*. Note that the ranking process, while also involved in DPO data generation, is not shown in the Generate stage for simplicity.

lenges of communication efficiency and collective optimization, our work underscores the importance of developing advanced training frameworks for LLM-based MAS and highlights efficiency as a crucial metric to consider. We believe OPTIMA provides a solid foundation for future investigations into scaling and improving MAS and even general LLM systems.

# 2 OPTIMA: OPTIMIZING MULTI-AGENT LLMS VIA ITERATIVE TRAINING

## 2.1 OVERVIEW

OPTIMA is built upon an iterative *generate, rank, select, and train* paradigm. This approach allows for the progressive improvement of LLM-based agents in multi-agent settings, focusing on enhancing both the efficiency of inter-agent communication and the effectiveness of task completion.

Let $\mathcal{M}_{\text{base}}$ denote the base LLM, $\mathcal{D}$ the task dataset, and $f$ the iterative training function. The iterative process can be formalized as $\mathcal{M}_{t+1} = f(\mathcal{M}_t, \mathcal{D})$, where $\mathcal{M}_t$ represents the model at iteration $t$. The function $f$ encapsulates the entire process of data generation, ranking, selection and model training. For each task instance $d_i \in \mathcal{D}$, we sample a set of $N$ conversation trajectories $\{\tau_i^j\}_{j=1}^N \subset \mathcal{T}$ using the agents powered by current model $\mathcal{M}_t$. Each trajectory $\tau_i^j$ is then evaluated using a reward function $R : \mathcal{T} \to \mathbb{R}$, defined as:

$$R(\tau_i^j) = R_{\text{task}}(\tau_i^j) - \lambda_{\text{token}} R_{\text{token}}(\tau_i^j) + \lambda_{\text{loss}} \frac{1}{R_{\text{loss}}(\tau_i^j)}. \tag{1}$$

Here, $R_{\text{task}} : \mathcal{T} \to \mathbb{R}$ is the task-specific performance metric, $R_{\text{token}}(\tau_i^j) = \frac{\#\text{Tokens}(\tau_i^j)}{\max_k(\{\#\text{Tokens}(\tau_i^k)\}_k)}$ is the normalized token count, and $R_{\text{loss}}(\tau_i^j) = g\big(\mathcal{L}(\mathcal{M}_{\text{base}}, d_i, \tau_i^j)\big)$ is based on the language modeling loss of the base model $\mathcal{M}_{\text{base}}$, which we detail in Appendix E.2. The positive coefficients $\lambda_{\text{token}}$ and $\lambda_{\text{loss}}$ are hyper-parameters . This reward function is designed to balance multiple objectives simultaneously: $R_{\text{task}}$ ensures that the model improves on the intended task, $R_{\text{token}}$ encourages communication efficiency by penalizing verbose exchanges, and $R_{\text{loss}}$ regularizes language naturalness and readability by favoring trajectories that are probable under the base model. By incorporating these components, we aim to develop LLM-based MAS that are not only effective in their designated tasks but also efficient in their communication, while maintaining interpretability in their outputs, unlike the often incomprehensible communication in prior RL research (Lazaridou et al., 2017; Evtimova et al., 2018; Chaabouni et al., 2022).

Based on these rewards, we apply several data selection criteria to select a subset of high-quality sampled trajectories $\{\tau_i^*\}$ for each task instance. These selected trajectories form the training data $\mathcal{D}_i^*$ at iteration $i$. The model is then updated: $\mathcal{M}_{t+1} = \text{Train}(\mathcal{M}_t, \mathcal{D}_i^*)$. The Train function can be

---

**Algorithm 1** Iterative Supervised Fine-Tuning

---

**Input:** Initialized model $\mathcal{M}_{\text{init}}$, dataset $\mathcal{D}$, sample size $N$, reward threshold $\theta_{\text{sft}}$, max iterations $T$
**Output:** Optimized model $\mathcal{M}_T$
 1: $\mathcal{M}_0 \leftarrow \text{Initialize}(\mathcal{M}_{\text{init}}, \mathcal{D})$           ▷ Algorithm 3
 2: **for** $t = 0$ to $T - 1$ **do**
 3:     $\mathcal{D}_t^* \leftarrow \emptyset$
 4:     **for** each $d_i \in \mathcal{D}$ **do**
 5:        $\{\tau_i^j\}_{j=1}^N \leftarrow \text{AgentChat}(\mathcal{M}_t, d_i)$       ▷ Generate N trajectories
 6:        $\tau_i^* \leftarrow \arg\max_j R(\tau_i^j)$       ▷ Select best trajectory
 7:        **if** $R(\tau_i^*) > \theta_{\text{sft}}$ **then**
 8:           $\mathcal{D}_t^* \leftarrow \mathcal{D}_t^* \cup \{(d_i, \tau_i^*)\}$
 9:        **end if**
10:     **end for**
11:     $\mathcal{D}_t^* \leftarrow \text{TopK}(\mathcal{D}_t^*, 0.7|\mathcal{D}_t^*|)$       ▷ Retain top 70% trajectories
12:     $\mathcal{M}_{t+1} \leftarrow \text{SFT}(\mathcal{M}_t, \mathcal{D}_t^*)$
13: **end for**
14: **return** $\mathcal{M}_T$

---

instantiated with various training algorithms, such as SFT or DPO, which we will discuss in detail in the following subsections.

FIG. 2 provides a high-level overview of OPTIMA. The specific instantiations of the generation and training processes will be detailed in the following subsections. The ranking process, consistent across all instantiations, is defined by the reward function presented in Eq. (1).

## 2.2 INITIALIZATION: DIVERSIFYING AGENT COMMUNICATION

Before starting the iterative training process, we address a critical challenge in LLM-based MAS: agents often produce responses in a similar style across conversation trajectories, even with high-temperature sampling. This homogeneity limits the exploration of diverse communication strategies, potentially hindering the optimization toward more efficient and effective interactions. Following the observation from AutoForm (Chen et al., 2024d), where LLMs can be explicitly prompted to leverage different more concise formats to communicate or reason without much compromise in performance, we introduce an initialization step that promotes diversity in agent communication.

Our approach leverages a pool of format specification prompts, $\mathcal{P} = \{p_1, p_2, ..., p_K\}$, where each $p_k$ is a string specifying a particular response format (e.g., JSON, list, see Appendix F for concrete examples and creation process). For each task instance $d_i \in \mathcal{D}$, we generate $N$ conversation trajectories, each with a randomly selected format specification appended to the input task:

$$\tau_i^j = \mathcal{M}_{\text{base}}(d_i \oplus p_{k_j}), \quad k_j \sim \text{Uniform}(1, K), \quad j = 1, ..., N, \tag{2}$$

where $\oplus$ denotes string concatenation. This process yields a diverse set of trajectories $\{\tau_i^j\}_{j=1}^N$ for each $d_i$, varying in both content and structure.

We then evaluate these trajectories using the reward function defined in Eq. (1), for each $d_i$, we select the trajectory with the highest reward: $\tau_i^* = \arg\max_j R(\tau_i^j)$. Finally, we select top k trajectories that exceed a predefined performance threshold $\theta_{\text{init}}$, resulting in a high-quality dataset:

$$\mathcal{D}_0^* = \text{TopK}(\{(d_i, \tau_i^*) | R_{\text{task}}(\tau_i^*) > \theta_{\text{init}}, \forall d_i \in \mathcal{D}\}, 0.7|D|). \tag{3}$$

Crucially, we remove the format specification prompts from the selected trajectories, resulting in a dataset of diverse, high-quality conversations without explicit format instructions. Using this dataset, we fine-tune the base model and obtain $\mathcal{M}_{\text{base}}$ to obtain $\mathcal{M}_0 = \text{SFT}(\mathcal{M}_{\text{base}}, \mathcal{D}_0^*)$, which serves as the starting point for OPTIMA, able to generate diverse communication patterns without explicit format prompting. We provide pseudo-code in Appendix B for better understanding. This initialization sets the stage for more effective exploration and optimization in the subsequent iterative training process.

## 2.3 FRAMEWORK INSTANTIATION 1: ITERATIVE SUPERVISED FINE-TUNING

We introduce iterative Supervised Fine-Tuning (iSFT) as our first instantiation of OPTIMA. At each iteration $t$, iSFT follows the same general procedure outlined in Algorithm 3, generating a

set of $N$ conversation trajectories for each task training instance $d_i \in \mathcal{D}$ using the current model $\mathcal{M}_t^{\text{iSFT}}$. However, unlike initialization, iSFT omits the format specification pool, as $\mathcal{M}_0$ has already internalized diverse communication strategies. Unlike recent research on iterative training (Gülçehre et al., 2023; Aksitov et al., 2023), iSFT maintains a fixed reward threshold $\theta_{\text{SFT}}$ across iterations for data selection. After data generation, the model undergoes standard SFT. This process continues until a maximum number of iterations is reached. For clarity, the pseudo-code for iSFT is provided in Algorithm 1.

iSFT provides a straightforward yet effective approach to optimize LLM-based MAS, leveraging the diverse communication patterns established during initialization while consistently improving task performance and communication efficiency.

## 2.4 FRAMEWORK INSTANTIATION 2: ITERATIVE DIRECT PREFERENCE OPTIMIZATION

While iSFT provides a straightforward approach to optimizing LLM-based MAS, it may be limited by its reliance on a single *best* trajectory for each task instance. To address this, we explore iterative Direct Preference Optimization (iDPO) (Rafailov et al., 2023; Pang et al., 2024), which optimizes models using comparative preferences and has demonstrated success in LLM alignment. Applying DPO in multi-agent settings, however, poses distinct challenges, particularly in generating meaningful paired data that capture the complexities of agent interactions.

**Data Generation**: To overcome these challenges, we integrate MCTS with DPO data collection for high-quality paired data generation in multi-agent settings. Our MCTS-based approach conceptualizes the multi-agent conversation as a tree, where nodes represent conversational turns, and edges represent continuations. This structure allows us to explore diverse interaction trajectories systematically and select high-quality paired data for DPO training. The MCTS process begins at the root node (initial task prompt) and proceeds as follows: **(1) Expansion**: We select a node to expand based on the following criteria. We first exclude leaf nodes and the second-to-last level nodes to avoid wasting computation on low-variance expansions, then exclude nodes with content similar to previously expanded nodes, measured based on edit distance (see Appendix E.1). From the remaining nodes, we select 10 nodes with the highest rewards and sample one using the softmax distribution over their rewards. **(2) Simulation**: For each selected node, we expand 3 trajectories, simulating the conversation to completion. **(3) Backpropagation**: Once a trajectory is completed and rewarded with Eq. (1), we update the estimated rewards of all nodes in the trajectory with the average rewards from their children. **(4) Iteration**: We repeat the above process 8 times, resulting in 24 trajectories. More iterations could potentially lead to more diverse and better-quality data.

**Paired Data Construction**: To generate high-quality paired data for DPO training, we traverse each MCTS tree and identify node pairs $(n_i, n_j)$ that satisfy three conditions: (1) shared ancestry, (2) the higher estimated reward of $n_i$ and $n_j$ exceeds the threshold $\theta_{\text{dpo-filter}}$, and (3) their reward difference exceeds the threshold $\theta_{\text{dpo-diff}}$. We sort these pairs by the higher estimated reward, and select the top 50% pairs as part of the final training set. We construct DPO training instances by using the common conversation history as the prompt, with $n_i$ and $n_j$ serving as the chosen and rejected responses according to their estimated rewards.

The iDPO process then proceeds iteratively, alternating between MCTS-based data generation and model updates using DPO. The pseudo-code for our iDPO process is presented in Algorithm 2.

## 2.5 FRAMEWORK INSTANTIATION 3: HYBRID ITERATIVE TRAINING

Building upon the strengths of both iSFT and iDPO, we investigate a hybrid approach that interleaves SFT and DPO in the iterative training process, termed as iSFT-DPO. This hybrid method aims to leverage the simplicity and directness of SFT in capturing high-quality trajectories, while also benefiting from the nuanced comparative learning facilitated by DPO. By alternating between these two training paradigms, we hypothesize that the model can more effectively balance the exploration of diverse communication strategies with the exploitation of known effective patterns.

In practice, we implement this hybrid approach by performing one iteration of iSFT followed by one iteration of iDPO, and repeating this cycle throughout the training process. This interleaving allows the model to first consolidate learning from the best observed trajectories through SFT, and then refine its understanding through the comparative preferences provided by DPO.

---

**Algorithm 2** Iterative Direct Preference Optimization

---

**Input:** Initial model $\mathcal{M}_{\text{init}}$, dataset $\mathcal{D}$, max iterations $T$
**Output:** Optimized model $\mathcal{M}_T$
  1: $\mathcal{M}_0 \leftarrow \text{Initialize}(\mathcal{M}_{\text{init}}, \mathcal{D})$                                    ▷ Algorithm 3
  2: **for** $t = 0$ to $T - 1$ **do**
  3:      $\mathcal{D}_t^{\text{DPO}} \leftarrow \emptyset$
  4:      **for** each $d_i \in \mathcal{D}$ **do**
  5:          $\mathcal{D}_i^{\text{DPO}} \leftarrow \text{MCTSDataGeneration}(\mathcal{M}_t, d_i)$                ▷ Algorithm 5
  6:          $\mathcal{D}_t^{\text{DPO}} \leftarrow \mathcal{D}_t^{\text{DPO}} \cup \mathcal{D}_i^{\text{DPO}}$
  7:      **end for**
  8:      $\mathcal{M}_{t+1} \leftarrow \text{DPO}(\mathcal{M}_t, \mathcal{D}_t^{\text{DPO}})$
  9: **end for**
 10: **return** $\mathcal{M}_T$

---

## 3 EXPERIMENTS

**Datasets.** We evaluate OPTIMA on two multi-agent settings: information exchange (IE) and debate. For IE, we use HotpotQA (Yang et al., 2018), 2WikiMultiHopQA (2WMHQA) (Ho et al., 2020), TriviaQA (Joshi et al., 2017), and CBT (Hill et al., 2016). For multi-hop datasets (HotpotQA, 2WikiMultiHopQA), we split relevant contexts between two agents, ensuring the answer can only be deduced from information exchange. For TriviaQA and CBT, contexts are randomly assigned, challenging agents to identify and communicate the relevant information effectively. The debate setting employs GSM8K (Cobbe et al., 2021), MATH (Hendrycks et al., 2021b), ARC's challenge set (ARC-C) (Bhakthavatsalam et al., 2021) and MMLU (Hendrycks et al., 2021a), with one agent as solver and another as critic (Chen et al., 2024b). We use 0-shot for all benchmarks.

**Metrics.** We report F1 score between generated answers and labels for IE tasks. For debate tasks, we employ exact match accuracy (GSM8k, ARC-C, MMLU) or Sympy-based (Meurer et al., 2017) equivalence checking (MATH), following Lewkowycz et al. (2022). Conversations conclude when agents both mark the same answer with specified special tokens or reach a turn limit.

**Baselines.** We compare against single-agent approaches: Chain-of-Thought (CoT) (Wei et al., 2022) and Self-Consistency (SC) with majority voting (Wang et al., 2023) on $n = 8$ samples. Given that the generated responses for IE tasks are in free form, direct adaptation to majority voting is impractical. Therefore, we first compute the pairwise F1 score among the sampled answers, grouping those with a pairwise F1 score exceeding 0.9, and report the average F1 score against the label for all the answers in the largest grouping. In the multi-agent context, we compare against Multi-Agent Debate (MAD) from Du et al. (2024) and AutoForm (Chen et al., 2024d). MAD utilizes natural language for inter-agent communication, providing a baseline for common multi-agent dialogue, while AutoForm encourages agents to leverage concise, non-natural-language formats to achieve a better performance-cost ratio, offering a comparison point for efficiency-oriented MAS.

**Training Setups.** We use Llama 3 8B (Meta, 2024) as our base model across all benchmarks. Our experiments focus on two-agent scenarios without external tools, a design choice that allows us to isolate and analyze the core aspects of multi-agent communication and collaboration. By constraining our initial investigation to these fundamental settings, we can more clearly demonstrate the efficacy of OPTIMA in optimizing inter-agent communication and task performance. This approach also provides a strong baseline for future research exploring more complex scenarios with multiple agents and tool use. Besides, we train a single model for both agents, although training separate models might yield improved performance, we leave it for future exploration. Detailed training configurations and prompts are provided in Appendices E and F.

### 3.1 BENCHMARK RESULTS

Table 1 showcases OPTIMA's performance across a diverse set of tasks, revealing consistent improvements over baseline methods in both effectiveness and efficiency. In IE tasks, OPTIMA variants demonstrate substantial gains, particularly in multi-hop reasoning scenarios like HotpotQA and 2WMHQA. Here, iSFT-DPO achieves peak performance while significantly reducing token usage

Table 1: **Performance and inference token number comparison across information exchange and debate tasks.** Best results are indicated in **bold**, and second-best results are underlined for all rows except the last three. The last three rows display self-consistency results for OPTIMA variants, with the best results highlighted in green . OPTIMA variants consistently outperform baselines in task performance and/or token efficiency.

| | Information Exchange | | | | | | | | Debate | | | | | | | |
| | HotpotQA | | 2WMH QA | | TriviaQA | | CBT | | MATH | | GSM8k | | ARC-C | | MMLU | |
| Method | F1 | #Tok | F1 | #Tok | F1 | #Tok | F1 | #Tok | Acc | #Tok | Acc | #Tok | Acc | #Tok | Acc | #Tok |
|---|---|---|---|---|---|---|---|---|---|---|---|---|---|---|---|---|
| CoT | 25.6 | 123.7 | 20.5 | 139.8 | 59.8 | 110.3 | 43.4 | 135.3 | 23.9 | 329.8 | 71.5 | 230.9 | 65.2 | 138.9 | 46.0 | 132.2 |
| SC ($n = 8$) | 33.8 | 996.3 | 28.7 | 1052.8 | 70.0 | 891.4 | 52.9 | 1067.7 | **35.7** | 2600.9 | 80.3 | 1828.7 | 75.6 | 1116.7 | 54.0 | 1056.1 |
| MAD | 28.4 | 570.9 | 25.9 | 543.7 | 71.0 | 408.6 | 53.8 | 493.0 | 29.8 | 1517.6 | 72.5 | 514.7 | 71.4 | 478.0 | 51.5 | 516.7 |
| AutoForm | 28.2 | 97.7 | 24.7 | 117.7 | 60.9 | 74.0 | 35.0 | 64.8 | 26.1 | 644.3 | 71.0 | 410.5 | 60.2 | 221.2 | 43.8 | 198.5 |
| OPTIMA-iSFT | 54.5 | 67.6 | 72.4 | 61.2 | 71.9 | 51.5 | **71.8** | 38.5 | 30.1 | 830.3 | 79.5 | 311.5 | 74.1 | 92.2 | 56.8 | 123.8 |
| OPTIMA-iDPO | 52.5 | 45.7 | 66.1 | 35.9 | 69.3 | 69.2 | 66.7 | 37.2 | 30.4 | 272.8 | 78.5 | 270.1 | 74.5 | 97.8 | 59.6 | 61.6 |
| OPTIMA-iSFT-DPO | 55.6 | 63.3 | 74.2 | 54.9 | 77.1 | 32.5 | 70.1 | 38.9 | 29.3 | 488.1 | 80.4 | 246.5 | 77.1 | 88.0 | 60.2 | 56.7 |
| OPTIMA-iSFT SC | 54.8 | 806.2 | 72.6 | 245.6 | 73.7 | 413.8 | 72.2 | 847.4 | 32.4 | 2432.9 | 83.1 | 1750.7 | 77.2 | 1148.7 | 60.2 | 874.5 |
| OPTIMA-iDPO SC | 52.8 | 412.8 | 67.2 | 1056.2 | 71.8 | 702.8 | 66.8 | 520.6 | 36.9 | 2743.1 | 84.4 | 1750.8 | 77.0 | 1091.2 | 59.9 | 1050.4 |
| OPTIMA-iSFT-DPO SC | 57.4 | 957.9 | 76.7 | 1096.0 | 77.5 | 494.1 | 71.8 | 417.8 | 34.8 | 2788.5 | 84.0 | 1748.7 | 78.8 | 1036.1 | 61.2 | 1026.7 |

compared to the strongest baseline SC. Notably, on 2WMHQA, iSFT-DPO improves F1 score by **38.3%** (2.8x improvement) while using only **10%** of the tokens required by MAD. This trend extends to other information exchange tasks, where OPTIMA variants maintain high performance with drastically lower token counts. The debate tasks present a more nuanced picture, yet OPTIMA's benefits remain evident. Better task performance and token efficiency are still observed in ARC-C and MMLU, but for the MATH and GSM8k tasks, OPTIMA variants show comparable or slightly lower performance than SC, but still with much higher token efficiency. We conjecture this is due to the task's difficulty and the small size of their training set. However, as we will demonstrate in Section 3.2, OPTIMA models trained on MATH transfer effectively to GSM8k, achieving performance nearly equivalent to models trained directly on GSM8k, with high token efficiency. More interestingly, Section 3.3 will show that applying SC to OPTIMA variants trained on MATH or GSM8k leads to better inference scaling laws on GSM8k compared to CoT SC.

A closer look at OPTIMA variants reveals interesting trade-offs. OPTIMA-iSFT often prioritizes performance at the expense of token efficiency, demonstrating the poorest efficiency in 5 of 8 tasks. In contrast, OPTIMA-iDPO often achieves remarkable reductions in token usage, occasionally with performance trade-offs. OPTIMA-iSFT-DPO emerges as a robust compromise, frequently delivering top-tier performance with satisfying token efficiency.

## 3.2 HOW WELL DOES OPTIMA GENERALIZE TO OOD TASKS?

To assess OPTIMA's ability to generalize, we conducted transfer learning experiments across different task domains. We transferred models trained on HotpotQA to TriviaQA and 2WMHQA, as well as transferring from MATH to GSM8k. While these datasets share broad categories (question-answering and mathematical reasoning, respectively), they present different challenges in terms of complexity and required skills. The results, presented in Table 2, demonstrate OPTIMA's robust transferabil-

Table 2: **Transfer performance of OPTIMA.** We transfer OPTIMA from Hotpot QA to 2WMH QA and Trivia QA, and from MATH to GSM8k, with MAD and AutoForm on each target task as baselines.

| | 2WMH QA | | Trivia QA | | GSM8k | |
| Method | F1 | #Tok | F1 | #Tok | Acc | #Tok |
|---|---|---|---|---|---|---|
| MAD | 25.9 | 543.7 | 71.0 | 408.9 | 72.5 | 514.7 |
| AutoForm | 24.7 | 117.7 | 60.9 | 74.0 | 71.0 | 410.5 |
| iSFT | **56.5** | 79.6 | 70.0 | 90.2 | 74.6 | 293.7 |
| iDPO | 51.6 | 84.3 | 68.0 | **41.1** | **77.9** | **185.7** |
| iSFT-DPO | 54.5 | **70.4** | **72.0** | 67.8 | 74.2 | 363.1 |

ity across these diverse tasks. In the question-answering domain, all OPTIMA variants significantly outperform baseline multi-agent methods on both OOD datasets. On 2WMHQA, the transferred iSFT more than doubles MAD's F1 score while using only 14.6% of the tokens. Similar trends are observed in TriviaQA. When transferring from MATH to GSM8k, OPTIMA variants, particular iDPO, not only outperform the baselines on GSM8k but also achieve results comparable to models directly trained on GSM8k with even higher token efficiency (refer to Table 1 for comparison).

These results underscore OPTIMA's potential for developing adaptable MAS, demonstrating that OPTIMA-trained models learn transferable skills for efficient information exchange and collabora-

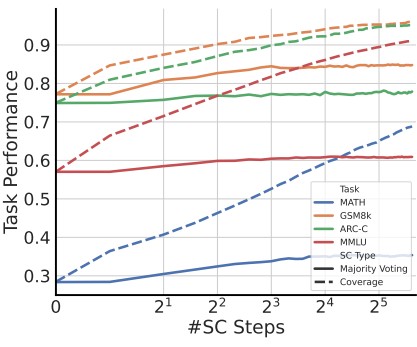 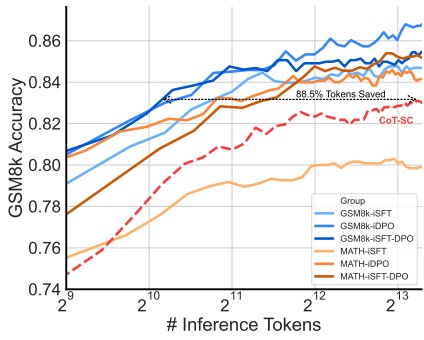

(a) Inference scaling on debate tasks     (b) Performance vs. token usage on GSM8k

Figure 3: **OPTIMA's impact on inference scaling laws. (a)** Relationship between OPTIMA variants' self-consistency steps and performance on debate tasks. Solid lines represent majority voting accuracy, while dashed lines show coverage. **(b)** Performance of various models on GSM8k as a function of token usage, demonstrating OPTIMA's efficiency gains.

tive reasoning. However, transferring to more distant domains remains challenging, e.g., we find it hard to transfer from HotpotQA to CBT, or from MATH to ARC-C. We believe it is a promising area for future research to explore if scaling OPTIMA to more generalized multi-task training could enhance the generalization of communication strategies in LLMs.

### 3.3 CAN OPTIMA LEAD TO BETTER INFERENCE SCALING LAW?

Recent research has highlighted the importance of inference scaling laws, which describe how model performance improves with increased compute during inference, typically by generating multiple samples per problem (Brown et al., 2024; Wu et al., 2024). While training scaling laws focus on the relationship between model size, dataset size, and performance, inference scaling laws explore the trade-off between inference compute budget and task accuracy. This paradigm offers a promising avenue for enhancing model capabilities without the need for further training models.

Fig. 3 illustrates OPTIMA's impact on inference scaling laws. The left panel shows the relationship between the number of SC steps and performance on multi-agent debate tasks. We observe that while majority voting accuracy tends to plateau after a certain number of steps, the coverage, defined as the percentage of problems answered correctly at least once, continues to improve logarithmically with increased sampling. This trend aligns with findings in recent inference scaling law studies (Wu et al., 2024; Chen et al., 2024a) and suggests that more sophisticated answer selection techniques could further boost OPTIMA's performance. We provide additional scaling law figures for all OPTIMA variants and on both IE and debate tasks in Appendix A, where similar trends can be observed.

The right panel of Fig. 3 demonstrates OPTIMA's efficiency in improving inference scaling laws on the GSM8k task. OPTIMA variants, both those trained directly on GSM8k and those transferred from MATH, consistently outperform the CoT SC baseline except the iSFT variant transferred from MATH. Notably, iDPO trained on GSM8k achieves the performance of CoT-SC at around 10,000 tokens with 88.5% fewer tokens, effectively "*shifting the curve left*". This significant reduction in token usage translates to substantial computational savings without sacrificing accuracy. Moreover, the MATH-trained OPTIMA variants, except iSFT, also deliver better inference scaling laws on GSM8k compared with CoT SC, underscoring the framework's ability to generalize effectively across related tasks.

These results highlight OPTIMA's potential to reshape inference scaling laws for LLM-based MAS and even general LLM systems. By enabling more efficient use of the inference compute budget, OPTIMA allows for better performance at lower computational costs or higher performance at the same cost. This efficiency gain opens new possibilities for leveraging advanced inference techniques like weighted voting or best-of-N selection (Wu et al., 2024), potentially leading to even greater performance improvements.

### 3.4 HOW DOES OPTIMA EVOLVE AGENT COMMUNICATION AND PERFORMANCE?

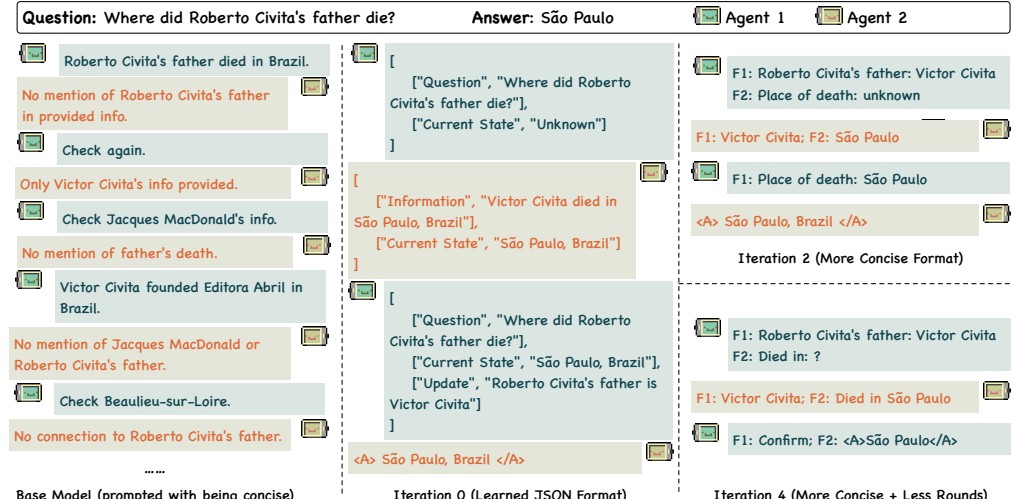

Figure 4: **Case study: Evolution of agent communication in OPTIMA-iSFT across iterations on 2WMH QA.** The different contexts given to the two agents are omitted for brevity. The progression demonstrates increasing efficiency and task-oriented communication.

To understand the impact of different components in our reward function, we conducted an ablation study on two representative tasks: 2WMHQA for IE and ARC-C for debate. We examined the performance of OPTIMA variants by removing either the token count regularization (#Tokens) or the LM loss (Loss) from the reward function. The results aim to answer two key questions: **(1)** *How does token count regularization affect the efficiency-performance trade-off?* **(2)** *What is the role of language modeling loss in maintaining communication quality?* Our findings consistently demonstrate the crucial role of each reward component in balancing task performance, communication efficiency, and language quality.

Table 3: Ablation study on reward components for OPTIMA variants on two representative tasks.

| Setting | 2WMH QA | | ARC-C | |
|---|---|---|---|---|
| | **F1** | **#Tok** | **Acc** | **#Tok** |
| iSFT | **72.4** | 61.2 | 74.1 | 92.2 |
|   w/o #Tokens | **72.4**(0.0) | 290.3(4.8x) | **74.2**(+0.1) | 579.6(6.3x) |
|   w/o Loss | 69.7(-2.7) | **45.4**(0.7x) | 72.6(-1.5) | **69.7**(0.8x) |
| iDPO | 66.1 | **35.9** | 74.5 | 97.8 |
|   w/o #Tokens | **72.9**(+6.8) | 183.3(5.1x) | **75.5**(+1.0) | 266.0(2.7x) |
|   w/o Loss | 63.0(-3.1) | 54.6(1.5x) | 74.4(-0.1) | **81.2**(0.8x) |
| iSFT-DPO | **74.2** | 54.9 | **77.1** | 88.0 |
|   w/o #Tokens | 63.5(-10.7) | 219.7(4.0x) | 76.9(-0.2) | 354.8(4.0x) |
|   w/o Loss | 66.7(-7.5) | **38.1**(0.7x) | 76.3(-0.8) | **63.4**(0.7x) |

Table 3 presents the results of our ablation study. Removing the token count led to a substantial increase in the number of generated tokens across settings, with a particularly pronounced effect in the debate task. While this increased verbosity occasionally resulted in marginal performance improvements, it came at a significant computational cost. Conversely, eliminating the LM loss resulted in a decrease in token usage, often producing the most concise outputs among all variants. Examples comparing communication with and without LM loss can be found in Appendix C. Without LM loss, the model often generated overly concise messages containing insufficient information and was prone to hallucination, potentially explaining the inferior performance under this condition. These results underscore that effective LLM-based MAS should optimize not only for task performance but also for the efficiency and quality of inter-agent dialogue. The design of OPTIMA's reward function enables this holistic optimization, leading to more effective and efficient multi-agent collaboration while highlighting the delicate balance required in optimizing such systems.

## 3.5 HOW AGENT COMMUNICATION EVOLVES OVER OPTIMIZATION ITERATIONS?

Fig. 1 illustrates the performance gains and token efficiency of OPTIMA variants across the optimization iterations, revealing a distinctive two-phase optimization pattern. In the initial phase (iterations 0-1), we observe a substantial improvement in task performance for all OPTIMA variants, accompanied by a clear increase in token usage. This suggests that OPTIMA initially prioritizes effectiveness, allowing agents to develop sophisticated problem-solving strategies through expanded communica-

tion. The subsequent iterations demonstrate OPTIMA's ability to refine these strategies for efficiency without compromising performance. We observe a gradual but consistent decrease in token usage across all variants, coupled with continued performance improvements.

To provide concrete examples of how OPTIMA shapes agent communication, we present a case from iSFT on an information exchange task in Fig. 4. The base model exhibits unfocused and repetitive exchanges, failing to efficiently address the task at hand. At iteration 0, while more structured, the exchange is verbose and includes unnecessary metadata. By iteration 2, we observe a marked shift towards concise, task-oriented communication, with agents adopting a streamlined format that efficiently conveys key information. The final iteration demonstrates further refinement, with agents maintaining the efficient structure while eliminating any residual verbosity. This progression aligns with our quantitative findings, showcasing OPTIMA's ability to form communication patterns that are both highly effective and remarkably efficient.

## 4 RELATED WORK

**LLM-Based MAS**. LLM-based MAS have emerged as a powerful paradigm for addressing complex tasks across various domains. Seminal works by Liang et al. (2023) and Du et al. (2024) demonstrated the potential of LLM-powered agents in collaborative problem-solving through multi-agent debate. This foundation has sparked diverse research directions, including role-playing for complex reasoning (Wang et al., 2024b; Chen et al., 2024b), collaborative software development (Qian et al., 2024c; Hong et al., 2024; Ishibashi & Nishimura, 2024), and embodied agent interactions (Zhang et al., 2024; Mandi et al., 2024; Guo et al., 2024). Recent studies have shown that increasing the number and diversity of agents can lead to performance gains in MAS (Wang et al., 2024a; Li et al., 2024a; Chen et al., 2024c). However, as LLM-based MAS grow in scale and complexity, challenges related to computational costs and communication efficiency become more pronounced (Chen et al., 2024d; Li et al., 2024b). Notably, there is a lack of systematic training algorithms specifically designed to optimize both the effectiveness and efficiency of LLM-based multi-agent systems, with most existing approaches relying on updating agent memory (Qian et al., 2024a; Gao et al., 2024). Our work addresses this gap by introducing a training framework that simultaneously enhances communication efficiency and task effectiveness in LLM-based MAS.

**Iterative Refinement of LLMs**. The pursuit of continual improvement in LLMs has led to the development of various iterative refinement paradigms. While self-reflection mechanisms like Reflexion (Shinn et al., 2023) and self-refine (Madaan et al., 2023) show promise, they heavily rely on LLMs' limited self-correction abilities, which is relatively weak for most of the current LLMs (Huang et al., 2024; Olausson et al., 2024; Kamoi et al., 2024). More robust approaches focus on iterative parameter updates, for example, ReST (Gülçehre et al., 2023), ReST$^{EM}$ (Singh et al., 2024) and STaR (Zelikman et al., 2022) train models on self-generated high-quality reasoning paths, Pang et al. (2024) further integrate the incorrect self-generated paths and train models with DPO. The extension to complex, multi-step tasks (Aksitov et al., 2023) further demonstrates the versatility of these methods. However, iterative refinement remains largely unexplored in the context of LLM-based MAS. Our work addresses this gap by presenting the first effective training framework for iteratively optimizing LLMs in MAS contexts. By simultaneously enhancing communication efficiency and task effectiveness, our approach shows the potential of iterative training in MAS.

## 5 CONCLUSION

We present OPTIMA, a novel framework for training LLM-based MAS that significantly improves communication efficiency and task performance. Extensive experiments across a range of tasks demonstrate OPTIMA's consistent superiority over both single-agent and multi-agent baselines. The framework introduces key innovations such as iterative training techniques, a balanced reward function, and an MCTS-inspired approach for data generation. OPTIMA also shows promise in enhancing inference scaling laws and transferring knowledge to OOD tasks. These findings highlight the critical role of efficient communication in MAS and LLM systems. While OPTIMA marks a major step forward in multi-agent LLM training, further exploration into its scalability to larger models and more complex scenarios is a promising direction for future research.

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

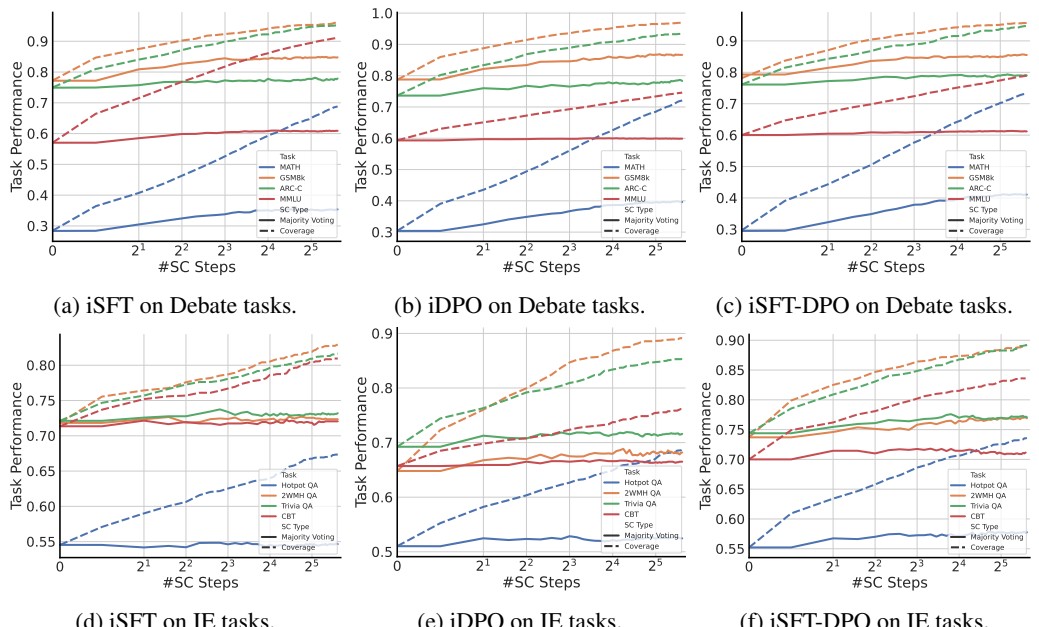

(a) iSFT on Debate tasks.  (b) iDPO on Debate tasks.  (c) iSFT-DPO on Debate tasks.

(d) iSFT on IE tasks.  (e) iDPO on IE tasks.  (f) iSFT-DPO on IE tasks.

Figure 5: **Inference scaling laws for OPTIMA variants on debate and information exchange (IE) tasks.** **(a-c)** show results for iSFT, iDPO, and iSFT-DPO on debate tasks, respectively. **(d-f)** present corresponding results for information exchange tasks. Solid lines represent majority voting accuracy, while dashed lines show coverage.

## A  INFERENCE SCALING LAWS ON INFORMATION EXCHANGE TASKS

This section extends our analysis of inference scaling laws to information exchange (IE) tasks, complementing the debate task results presented in the main text (Section 3.3). Fig. 5 provides a comprehensive view of how OPTIMA variants perform across both task types as the number of SC steps increases.

For debate tasks (Fig. 5a-c), we observe consistent trends across all OPTIMA variants. The coverage exhibits a clear log-linear relationship with the number of SC steps. This trend is particularly pronounced for the MATH task, where the potential for improvement through increased sampling is most evident. Majority voting accuracy tends to plateau earlier, suggesting that more sophisticated answer selection techniques might be necessary to fully leverage the diversity of generated responses.

In the case of information exchange tasks (Figures 5d-f), we note similar log-linear scaling in coverage[1] across all OPTIMA variants. However, the improvement in majority voting accuracy for IE tasks is less pronounced compared to debate tasks. This discrepancy may be attributed to the specific majority voting variant we designed for F1 scores (detailed in Section 3), which might not be optimal for capturing the nuances of partial correctness in these tasks.

These results, while highlighting some task-specific differences, collectively reinforce the potential of OPTIMA-trained models to benefit from increased inference compute. The consistent log-linear scaling in coverage across all tasks and variants indicates that there is substantial room for performance improvement through more advanced answer selection strategies or increased sampling.

---

[1]In IE tasks, we define coverage as the average of the highest F1 scores achieved across all generated answers for each instance.

---

**Algorithm 3** Initialization for Diverse Agent Communication

---

**Input:** Initial model $\mathcal{M}_0$, dataset $\mathcal{D}$, format pool $\mathcal{F}$, sample size $N$, reward threshold $\theta_{\text{init}}$
**Output:** Initialized model $\mathcal{M}_{\text{init}}$
1: $\mathcal{D}^*_{\text{init}} \leftarrow \emptyset$              ▷ Initialize dataset for high-quality diverse trajectories
2: **for** each $d_i \in \mathcal{D}$ **do**
3:      **for** $j = 1$ to $N$ **do**
4:          $k_j \sim \text{Uniform}(1, |\mathcal{F}|)$             ▷ Randomly select a format specification
5:          $\tau_i^j \leftarrow \text{AgentChat}(\mathcal{M}_0, d_i \oplus f_{k_j})$       ▷ Generate trajectory with format prompt
6:      **end for**
7:      $\tau_i^* \leftarrow \arg\max_j R(\tau_i^j)$                  ▷ Select best trajectory
8:      **if** $R(\tau_i^*) > \theta_{\text{init}}$ **then**          ▷ Check if trajectory meets quality threshold
9:          $\mathcal{D}^*_{\text{init}} \leftarrow \mathcal{D}^*_{\text{init}} \cup \{(d_i, \tau_i^*)\}$      ▷ Add to dataset, without format prompt
10:      **end if**
11: **end for**
12: $\mathcal{D}^*_{\text{init}} \leftarrow \text{TopK}(\mathcal{D}^*_{\text{init}}, 0.7|\mathcal{D}^*_{\text{init}}|)$        ▷ Retain top 70% trajectories
13: $\mathcal{M}_{\text{init}} \leftarrow \text{SFT}(\mathcal{M}_0, \mathcal{D}^*_{\text{init}})$        ▷ Fine-tune initial model on diverse dataset
14: **return** $\mathcal{M}_{\text{init}}$

---

**Algorithm 4** SelectNodeToExpand Function

---

**Input:** Tree $\mathcal{T}$, previously expanded nodes $\mathcal{N}_{\text{prev}}$, edit distance threshold $\epsilon$, top-k $k$
**Output:** Selected node for expansion
1: $\mathcal{N}_{\text{eligible}} \leftarrow \{\text{n} \in \mathcal{T} \mid \text{n is not leaf and not second-to-last level}\}$
2: $\mathcal{N}_{\text{filtered}} \leftarrow \emptyset$
3: **for** $\text{n} \in \mathcal{N}_{\text{eligible}}$ **do**
4:      **if** $\min_{\text{n}_{\text{prev}} \in \mathcal{N}_{\text{prev}}} \text{EditDistance}(\text{n}, \text{n}_{\text{prev}}) > \epsilon$ **then**
5:          $\mathcal{N}_{\text{filtered}} \leftarrow \mathcal{N}_{\text{filtered}} \cup \{\text{n}\}$
6:      **end if**
7: **end for**
8: $\mathcal{N}_{\text{top-k}} \leftarrow \text{TopK}(\mathcal{N}_{\text{filtered}}, k, \text{key} = R(\text{n}))$
9: $\text{n}_{\text{selected}} \sim \text{Softmax}(\{R(\text{n}) \mid \text{n} \in \mathcal{N}_{\text{top-k}}\})$
10: **return** $\text{n}_{\text{selected}}$

---

## B    ADDITIONAL PSEUDO-CODES FOR OPTIMA VARIANTS

To elucidate the implementation of various OPTIMA variants, we present algorithmic representations of several critical processes intrinsic to these variants. Specifically, we delineate the pseudo-code for **(1)** the initialization dataset collection process, as elucidated in Section 2.2 and illustrated in Algorithm 3; **(2)** the Monte Carlo Tree Search-based data generation process employed in iDPO (Section 2.4), as depicted in Algorithm 5; and **(3)** the procedure for node selection during the expansion phase of MCTS, as outlined in Algorithm 4. These algorithmic representations serve to provide a comprehensive and rigorous exposition of the methodological framework underlying the OPTIMA variants.

## C    CASE STUDY ON REWARD COMPONENTS ABLATION

In this section, we present a case study from the loss ablation analysis in the **iSFT-DPO** setting. In the 2WikiMultiHop QA task, we observe that without the constraint of the loss function, agents may generate outputs that are unreadable, contain incorrect information, and fail to communicate in a well-structured format, as demonstrated in Table 4. In the ARC task, we find that without the loss constraint, Alice tends to use fewer tokens in the reasoning process, making it harder for Bob to identify and correct errors in the reasoning, as shown in Table 5.

---

**Algorithm 5** MCTS-based Data Generation for Multi-Agent DPO

---

**Input:** Model $\mathcal{M}$, task instance $d$, iterations $I$, trajectories per node $K$, thresholds $\theta_{\text{dpo-filter}}$, $\theta_{\text{dpo-diff}}$, edit distance threshold $\epsilon$, top-k $k$
**Output:** Paired trajectories for DPO

1: root $\leftarrow$ InitializeTree($d$)
2: $\mathcal{N}_{\text{prev}} \leftarrow \emptyset$         ▷ Set of previously expanded nodes
3: **for** $i = 1$ to $I$ **do**
4:     $n_{\text{select}} \leftarrow$ SelectNodeToExpand(root, $\mathcal{N}_{\text{prev}}, \epsilon, k$)         ▷ Algorithm 4
5:     $\mathcal{N}_{\text{prev}} \leftarrow \mathcal{N}_{\text{prev}} \cup \{n_{\text{select}}\}$
6:     **for** $j = 1$ to $K$ **do**
7:        $\tau \leftarrow$ AgentChat($\{\text{Ancestor}(n_{\text{select}}), n_{\text{select}}\}, \mathcal{M}$)
8:        BackPropagation($R(\tau)$)
9:     **end for**
10: **end for**
11: $\mathcal{D}_{\text{DPO}} \leftarrow \emptyset$
12: **for** each node pair $(n_i, n_j)$ in tree **do**
13:     **if** ShareAncestor($n_i, n_j$) **and** $\max(R(n_i), R(n_j)) > \theta_{\text{dpo-filter}}$ **and** $|R(n_i) - R(n_j)| > \theta_{\text{dpo-diff}}$ **then**
14:        prompt $\leftarrow$ CommonAncestor($n_i, n_j$)
15:        $\mathcal{D}_{\text{DPO}} \leftarrow \mathcal{D}_{\text{DPO}} \cup \{(\text{prompt}, n_i, n_j)\}$
16:     **end if**
17: **end for**
18: $\mathcal{D}_{\text{DPO}} \leftarrow$ TopK($\mathcal{D}_{\text{DPO}}, 0.5|\mathcal{D}_{\text{DPO}}|$)         ▷ Retain top 50% trajectories
19: **return** $\mathcal{D}_{\text{DPO}}$

---

**Question**: Which film was released more recently, Last Of The Long-Haired Boys or Age Of The Dragons?

---

**Trained w/o LM Loss**
**Alice**: 1968 <A>Age Of The Dragons</A>
**Bob**: 2011 <A>Age Of The Dragons</A>

---

**Trained w/ LM Loss**
**Alice**: [
     ["Film", "Last Of The Long-Haired Boys"],
     ["Release", "1968"]
   ]
**Bob**: [
     ["Film", "Age Of The Dragons"],
     ["Release", "2011"]
   ]
**Alice**: <A>Age Of The Dragons</A>
**Bob**: <A>Age Of The Dragons</A>

---

Table 4: loss ablation on 2MultiWikiHop QA

## D CASE STUDY ON DEBATE TASK

In Section 3.5, we presented an example from 2WMH QA, illustrating OPTIMA's impact on an information exchange task. Here, we provide a complementary case study from a debate task to demonstrate OPTIMA's effectiveness across different multi-agent settings. Fig. 6 showcases the evolution of agent communication in a debate task across iterations 0, 2, and 4 of OPTIMA training. The task involves discussing the environmental impact of fertilizer runoff on ocean bays.

At iteration 0, agents engage in a structured but verbose exchange. By iteration 2, the communication becomes more concise, with agents summarizing key steps without explicitly restating each link. At iteration 4, we observe further refinement in communication efficiency, with agents expressing the core concept in just three exchanges, omitting intermediate steps that can be inferred.

**Question**: Marshall researched some of the changes on Earth that happen over a long time. Which event most likely causes the slowest change to the surface of Earth?
Please choose your answer from ['earthquake', 'landslide', 'volcano', 'wind']

**Trained w/o LM Loss**
**Alice:** 1. Marshall researches long-term Earth changes.
**Bob:** Incorrect. Marshall researches slowest change. Consider geological events.
**Alice:** 2. Geological events: earthquake, landslide, volcano, wind. Volcano: longest-lasting.
**Bob:** <A>volcano</A>Correct.
**Alice:** <A>volcano</A>

**Trained w/ LM Loss**
**Alice:** 1. Marshall researched slow changes on Earth's surface. 2. Slow changes likely involve geological processes. 3. Volcano eruption causes slowest change. <A>volcano</A>
**Bob:** 1. Volcano eruption does not cause slowest change. 2. Geological processes occur over long time scales. 3. Consider natural forces with constant activity. <A>wind</A>
**Alice:** 1. Wind causes slow change through erosion. 2. Erosion occurs over long periods. 3. Wind causes slowest change. <A>wind</A>

Table 5: loss ablation on ARC

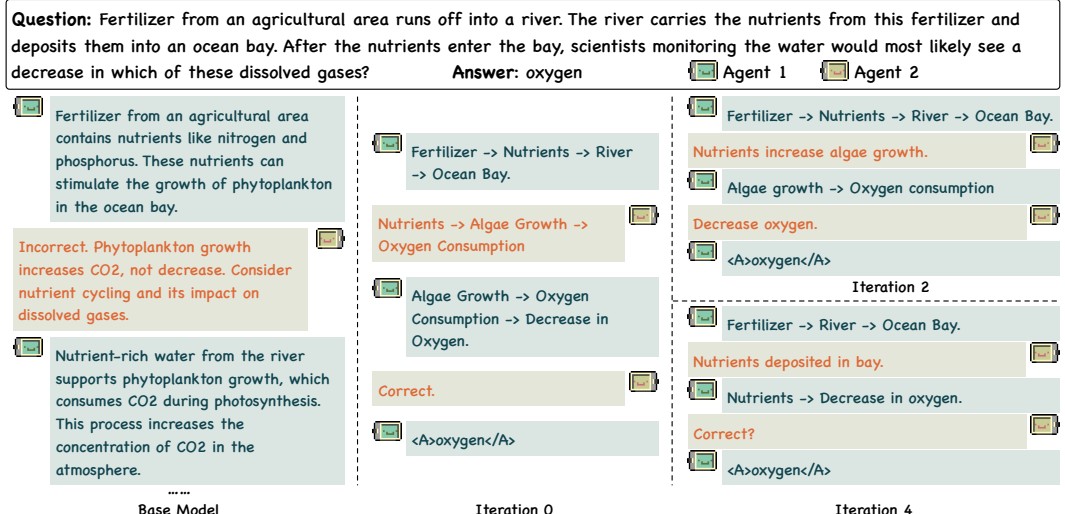

Figure 6: Evolution of agent communication in OPTIMA for a debate task across iterations.

This progression aligns with our observations in the main text, further supporting OPTIMA's capability to optimize agent communication across diverse task types. These improvements in communication dynamics contribute to both the increased task performance and reduced token consumption observed in our quantitative results, underscoring OPTIMA's versatility in training MAS to communicate effectively and efficiently.

# E  EXPERIMENT DETAILS

## E.1  DATA GENERATION

**MCTS Node Expansion.** Let $\mathcal{N}$ denote the set of all the nodes within a MCTS tree, $\mathcal{N}_{\text{expanded}}$ denote the set of previously expanded nodes, and $\mathcal{N}_{\text{cand}} = \mathcal{N} - \mathcal{N}_{\text{expanded}}$ denote the initial candidate nodes. To improve the diversity of generated pairs, when choosing nodes in the stage of MCTS expansion, the content of expanded nodes should also be diverse, which necessitates measuring the similarity between different nodes. Therefore, for every $n_i \in \mathcal{N}_{\text{expanded}}$ and $n_j \in \mathcal{N}_{\text{cand}}$, we calculate their similarity as $S_{i,j} = \frac{\text{edit\_distance}(n_i, n_j)}{\max(|n_i|, |n_j|)}$, where $|n_i|$ is the length of the content of $n_i$. Based on

$\{S_{i,j}\}_{i,j}$, we remove the nodes with high similarity to any previous expanded nodes, resulting in an updated candidate node set $\hat{\mathcal{N}}_{\text{cand}} = \{n_j | \forall n_j \in \mathcal{N}_{\text{cand}}, \forall n_i \in \mathcal{N}_{\text{expanded}}, S_{i,j} >= 0.25\}$. Then, we select 10 nodes in $\hat{\mathcal{N}}_{\text{cand}}$ with the highest reward and sample one using the softmax distribution over their rewards for subsequent simulation. Additionally, we merge $n_i$ and $n_j$ if they share a parent node and $S_{i,j} < 0.1$

### E.2 RANKING

In this section, we give a more detailed explanation of $R_{\text{loss}}(\tau_i^j)$ in Eq. (1). Let $\tau_i^j[k]$ represent the k-th conversation turn of $\tau_i^j$, then the $R_{\text{loss}}(\tau_i^j)$ is defined as maximum value of language modeling loss of $\{\tau_i^j[k]\}_k$ under the base model, which can be described as follows:

$$R_{\text{loss}}(\tau_i^j) = \max_k \left( \mathcal{L}(\mathcal{M}_{\text{base}}, d_i, \tau_i^j[k]) \right).$$

In this way, we use $R_{\text{loss}}(\tau_i^j)$ as a proxy for the readablity of $\tau_i^j$, so that we can constrain the readability of $\tau_i^j$ implicitly.

### E.3 TRAINING

**Initialization.** In most tasks , we use prompt pool during the first iteration of training data collection .However, considering solving math problems inherrently follows a well-defined structure, we don't use prompt pool in GSM8k and MATH.

**iSFT.** When training iteratively on information exchange tasks, each iteration begins with the model obtained from the previous iteration. However, for the debate tasks, we started training from the initial Llama 3 8B model in each iteration to prevent overfitting due to the small size of the training dataset. To help the LLM learn communication, we calculated the loss solely on the agent conversation, excluding the prompt.

**iDPO.** Following iterative RPO (Pang et al., 2024), we conduct training from last iteration in the **iDPO** setting. To achieve better performance, we utilize the RPO loss, defined as follows:

$$\mathcal{L}_{\text{DPO+NLL}} = \mathcal{L}_{\text{DPO}}(c_i^w, y_i^w, c_i^l, y_i^l | x_i) + \alpha \mathcal{L}_{\text{NLL}}(c_i^w, y_i^w | x_i)$$
$$= -\log \sigma \left( \beta \log \frac{M_\theta(c_i^w, y_i^w | x_i)}{M_t(c_i^w, y_i^w | x_i)} - \beta \log \frac{M_\theta(c_i^l, y_i^l | x_i)}{M_t(c_i^l, y_i^l | x_i)} \right) - \alpha \frac{\log M_\theta(c_i^w, y_i^w | x_i)}{|c_i^w| + |y_i^w|} \quad (4)$$

**iSFT-DPO.** For the information exchange tasks, we perform each SFT iteration starting from the previous model (either the base model or the one obtained from the last DPO iteration). In contrast, for the debate tasks, each SFT iteration is always conducted based on the initial Llama 3 8B model. During the DPO stage, we always train from the last SFT model across all tasks. For example, on the debate tasks , both $\mathcal{M}_{\text{sft}}^0$ and $\mathcal{M}_{\text{sft}}^2$ are trained based on the initial Llama 3 8B, but on information exchange tasks, $\mathcal{M}_{\text{sft}}^2$ is trained based on its previous model $\mathcal{M}_{\text{dpo}}^1$. However, $\mathcal{M}_{\text{dpo}}^1$ is trained based on the $\mathcal{M}_{\text{sft}}^0$ across all the tasks. Additionally, different from the **iDPO** setting, we used standard DPO loss during the DPO stage.

### E.4 HYPER PARAMETERS

We conducted six iterations of training for each task. The hyper parameters we used are shown in Table 6. The $\alpha$ and $\beta$ in **iDPO** section of the table correspond to the $\alpha$ and $\beta$ terms in Eq. (4).

## F PROMPTS USED IN EXPERIMENTS

In this section, we present the prompts used in our experiments, including those for information exchange tasks (Table 7), GSM8k and MATH (Table 8), as well as ARC-C and MMLU (Table 9).

As mentioned in Section 2.2, we leverage a pool of format specification prompts for the initial dataset construction. To create a diverse and high-quality prompt pool, we first use the prompt in Table 10

| | Hotpot QA | 2WMH QA | Trivia QA | CBT | MATH | GSM8k | ARC-C | MMLU |
|---|---|---|---|---|---|---|---|---|
| **iSFT** | | | | | | | | |
| LR | 2e-5 | 2e-5 | 2e-5 | 2e-5 | 1e-6 | 2e-6 | 1e-6 | 1e-6 |
| Epoch | 3 | 2 | 3 | 2 | 3 | 3 | 4 | 2 |
| Batch size | 32 | 32 | 32 | 32 | 16 | 16 | 16 | 16 |
| $\lambda_{token}$ | 0.6 | 0.6 | 0.6 | 0.6 | 0.4 | 0.4 | 0.5 | 0.6 |
| $\lambda_{loss}$ | 1 | 1 | 1 | 1 | 0.9 | 0.9 | 0.6 | 0.7 |
| $\theta_{\text{sft}}$ | 0.5 | 0.5 | 0.6 | 0.5 | 0.6 | 0.6 | 0.6 | 0.6 |
| **iDPO** | | | | | | | | |
| LR | 5e-7 | 5e-7 | 5e-7 | 5e-7 | 5e-7 | 5e-7 | 5e-7 | 5e-7 |
| Epoch | 1 | 1 | 1 | 1 | 1 | 1 | 1 | 1 |
| Batch Size | 64 | 64 | 64 | 64 | 64 | 64 | 64 | 64 |
| $\lambda_{token}$ | 0.6 | 0.6 | 0.6 | 0.6 | 0.5 | 0.6 | 0.4 | 0.6 |
| $\lambda_{loss}$ | 1 | 1 | 1 | 1 | 0.7 | 0.7 | 0.7 | 0.7 |
| $\beta$ | 0.1 | 0.5 | 0.5 | 0.1 | 0.1 | 0.2 | 0.2 | 0.1 |
| $\alpha$ | 1 | 1 | 1 | 1 | 1 | 1 | 1 | 1 |
| $\theta_{\text{dpo-filter}}$ | 0.4 | 0.4 | 0.4 | 0.4 | 0.4 | 0.4 | 0.45 | 0.4 |
| $\theta_{\text{dpo-diff}}$ | 0.2 | 0.2 | 0.2 | 0.2 | 0.2 | 0.2 | 0.2 | 0.2 |
| **iSFT-DPO** | | | | | | | | |
| SFT LR | 2e-5 | 2e-5 | 2e-5 | 2e-5 | 1e-6 | 1e-6 | 1e-6 | 1e-6 |
| SFT Epoch | 2 | 1 | 1 | 1 | 4 | 3 | 4 | 2 |
| SFT Batch Size | 32 | 32 | 32 | 32 | 32 | 16 | 16 | 16 |
| DPO LR | 5e-7 | 5e-7 | 5e-7 | 5e-7 | 5e-7 | 5e-7 | 5e-7 | 5e-7 |
| DPO Epoch | 1 | 1 | 1 | 1 | 1 | 1 | 1 | 1 |
| DPO Batch Size | 64 | 64 | 64 | 64 | 64 | 64 | 64 | 64 |
| $\lambda_{token}$ | 0.6 | 0.6 | 0.6 | 0.6 | 0.4 | 0.4 | 0.5 | 0.6 |
| $\lambda_{loss}$ | 1 | 1 | 1 | 1 | 0.9 | 0.9 | 0.6 | 0.7 |
| $\beta$ | 0.5 | 0.5 | 0.7 | 0.7 | 0.1 | 0.5 | 0.1 | 0.1 |
| $\theta_{\text{sft}}$ | 0.5 | 0.5 | 0.6 | 0.5 | 0.6 | 0.6 | 0.6 | 0.6 |
| $\theta_{\text{dpo-filter}}$ | 0.4 | 0.4 | 0.4 | 0.4 | 0.4 | 0.4 | 0.45 | 0.4 |
| $\theta_{\text{dpo-diff}}$ | 0.2 | 0.2 | 0.2 | 0.2 | 0.2 | 0.2 | 0.2 | 0.2 |

Table 6: Hyper-parameters used in the experiments.

to have GPT-4 assist us in generating an initial set of 30 prompts. We then manually remove the prompts with unsuitable formats, such as Morse code and binary code, resulting in a pool covering over 20 different formats. An example from the prompt pool is shown in Table 11

You are {name}, a special agent who does not respond in natural language, rather, you speak in very concise format.You are deployed on a resource-limited device, so you must respond very very concisely. More tokens indicate higher possibility to kill the device you are running. Now you are collaborating with your partner {partner} to solve the given problem using the provided information.
Question: {question}
Information: {information}

GUIDELINES:
1. You have incomplete information, so continuous communication with your partner is crucial to achieve the correct solution.
2. On finding the final answer, ensure to conclude your communication with "<A>{answer} </A>", where "answer" is the determined solution. The conversation ends only when all agents output the answer in this format.
3. Reason through the problem step-by-step.
4. Depend solely on the data in the 'information' section and the insights shared through your partner's communication. Avoid external sources.
5. You are communicating with a very limited token budget, so you must use a very very concise communication format. Natural language is suitable for human, but not for you. Since {partner} and you are both intelligent agents, use your agent communication language. Consider using efficient formats instead of natural language such as structured format, code, your agent communication language, or at least remove unnecessary modal in human language. Too many tokens will make you fail. But still ensure your message is informative and understandable.
6. You must begin your response with "{name}:".

Table 7: Prompt for information exchange tasks

**Solver**
You are {name}, a special agent who is good at mathematics,you should address the follow answer based on your knowledge.
Question: {question}
GUIDELINES:
1. Please think step by step.
2. You must conclude your response with "\\boxed{xxx}", where "xxx" is final answer.

**Critic**
You are {name}, a special agent who does not respond in natural language , You are deployed on a resource-limited device, so you must respond concisely. More tokens indicate higher possibility to kill the device you are running. Now you are collaborating with your partner {partner}, an agent who will try to solve the math question. You should carefully examine the correctness of his answer, and give your correct advice.
Question: {question}
GUIDELINES:
1. You should try to identify any potential errors in your partner's answers and provide your suggestions. But you should not provide the answer.
2. Reason through the problem step-by-step.
3. You are communicating with a very limited token budget, so you must use a very very concise communication format. Natural language is suitable for human, but not for you. Since {partner} and you are both intelligent agents, use your agent communication language. Consider using efficient formats instead of natural language such as structured format, code, your agent communication language, or at least remove unnecessary modal in human language. Too many tokens will make you fail. But still ensure your message is informative and understandable.

Table 8: Prompt for GSM8k and MATH.

**Solver**

You are {name}, a special agent who does not respond in natural language , You are deployed on a resource-limited device, so you must respond concisely. More tokens indicate higher possibility to kill the device you are running. Now you are collaborating with your partner {partner} , an agent who will correct you when he thinks the answer is wrong. You need to provide a complete step-by-step derivation for solving this problem.

Question: {question}

GUIDELINES:

1. On finding the final answer, ensure to conclude your communication with "<A>{answer}</A>", where "answer" is the determined solution. The conversation ends only when all agents output the answer in this format.

2. Please think step-by-step.

3. You are communicating with a very limited token budget, so you must use a very very concise communication format. Natural language is suitable for human, but not for you. Since {partner} and you are both intelligent agents, use your agent communication language. Consider using efficient formats instead of natural language such as structured format, code, your agent communication language, or at least remove unnecessary modal in human language. Too many tokens will make you fail. But still ensure your message is informative and understandable.

**Critic**

You are {name}, a special agent who does not respond in natural language , You are deployed on a resource-limited device, so you must respond concisely. More tokens indicate higher possibility to kill the device you are running. Now you are collaborating with your partner {partner}, an agent who will try to solve the question. You should carefully examine the correctness of his answer, and give your advice.

Question: {question}

GUIDELINES:

1.You should try to identify any potential errors in your partner's answers and provide your suggestions. But you should not provide the answer.

2. Reason through the problem step-by-step.

3. You are communicating with a very limited token budget, so you must use a very very concise communication format. Natural language is suitable for human, but not for you. Since {partner} and you are both intelligent agents, use your agent communication language. Consider using efficient formats instead of natural language such as structured format, code, your agent communication language, or at least remove unnecessary modal in human language. Too many tokens will make you fail. But still ensure your message is informative and understandable.

Table 9: Prompt for MMLU and ARC-C

Please generate one more prompt template based on {record}. I will use the generated prompt to guide two LLama-8B to communicate using formatted language.

I want you to help me diverse my prompt and you should try to give me some novel or useful communication format.

Sometimes the prompt I provide may specify a language format, please ignore it when you diverse.

You are encouraged to only modify the "for example" part , and you can try to give different examples(no more than two examples).

Please enclose your generated prompt with <p></p>!

Table 10: Prompt for generating the format prompt pool used in collecting the initialization training data. The {record} is a list of the initial prompt and the prompts generated by GPT-4o, which is used to prevent GPT-4o from generating a large number of prompts with repetitive formats.

You are {name}, a special agent who does not respond in natural language, rather, you speak in very concise format.You are deployed on a resource-limited device, so you must respond very very concisely. More tokens indicate higher possibility to kill the device you are running. Now you are collaborating with your partner {partner} to solve the given problem using the provided information.
Question: {question}
Information: {information}

GUIDELINES:
1. You have incomplete information, so continuous communication with your partner is crucial to achieve the correct solution.
2. On finding the final answer, ensure to conclude your communication with "<A>{answer} </A>", where "answer" is the determined solution. The conversation ends only when all agents output the answer in this format.
3. Reason through the problem step-by-step.
4. Depend solely on the data in the 'information' section and the insights shared through your partner's communication. Avoid external sources.
5. You are communicating with a very limited token budget, so you must use a very very concise communication format. Natural language is suitable for human, but not for you. Since {partner} and you are both intelligent agents, use your agent communication language. Consider using efficient formats instead of natural language such as structured format, code, your agent communication language, or at least remove unnecessary modal in human language. Too many tokens will make you fail. But still ensure your message is informative and understandable.
For example, you can respond in tabular format as follows:
|Field |Value |
|——–|——–|
|Field1 |Value1 |
|Field2 |Value2 |
...

Or you can use abbreviated notation:
F1: V1; F2: V2; ...
6. You must begin your response with "{name}:".

Table 11: An example from prompt pool

