# OpenReview forum: "Optima: Optimizing Effectiveness and Efficiency for LLM-Based Multi-Agent System"
_ICLR.cc/2025/Conference — Submitted to ICLR 2025_

### Official Review · Reviewer_uBkL · 2024-10-22

**Soundness:** 3
**Presentation:** 3
**Contribution:** 3
**Rating:** 6
**Confidence:** 4

**Summary:**

The paper introduces OPTIMA, a novel framework designed to optimize Large Language Model (LLM)-based multi-agent systems (MAS), which shows the potential towards scalable, efficient, and effective LLM-based MAS.. The key focus is on enhancing communication efficiency and task effectiveness through an iterative generate, rank, select, and train paradigm. OPTIMA utilizes a hybrid of reinforcement learning (RL) techniques, including Supervised Fine-Tuning (SFT) and Direct Preference Optimization (DPO), along with Monte Carlo Tree Search (MCTS)-inspired data generation. The framework is evaluated on two multi-agent settings: (a) information exchange, including information-asymmetric question answering and  (b) debate, encompassing mathematical and reasoning tasks. OPTIMA’s efficiency gains open new possibilities for leveraging inference-compute more effectively, potentially leading to improved inference-time scaling laws.

**Strengths:**

1, Clarity: The paper is well written and have full proof for their argument, the concepts in this paper are easy to follow and understand.
2, Quality: The performance of their designed pipeline is impressive and outperform the baseline a lot.
3, Originality: The paper introduced a new way to enable the development of MAS that are not only effective and efficient but also maintain interpretable communication patterns, which solves the core issues of communication efficiency and collective optimization.

**Weaknesses:**

1, The method is only evaluated on two tasks: (a) information exchange, including information-asymmetric question answering and  (b) debate, encompassing mathematical and reasoning tasks.  Although the performance of proposed method is great on those two tasks, but not sure how this method will perform on the other more complex tasks such as: StarCraft II, Hanabi and so on. (those two environment are broadly used in reinforcement learning tasks and there are also works of using LLM to play those two games)

2, The scalability of this method. The number of agents in showed tasks are limited,. And because the proposed method are aimed to solve the problem of multi-agent tasks, using more agents in environment(maybe at least 4 or 5 agents?) will be more convincible.

3, They use Llama 3 8B as their base model across all benchmarks. However, can this method still have significant improvement compared to baseline if using other base models still remains as a question and need further proof.

**Questions:**

The method proposed in this paper is great, but still I have following concerns which may need further experiments:
1, More tasks study will make this proposed method more convincible, including using more complex tasks and more agents in a task. (StarCraft II, Hanabi and any other tasks which are broadly used in using LLM in MAS problems)

2, Try at least one more base model on all the tasks such as Llama 3.2 or Llama 3 70B

---

> ### Author Response · Authors · 2024-11-20
>
> We appreciate your thoughtful feedback and for recognizing the contributions of our work. Your comments and suggestions have helped us identify areas for further clarification and improvement. Below, we address each of your concerns in detail.
>
> ---
>
> **1. Evaluation on Additional Tasks (StarCraft II, Hanabi, etc.)**
>
> Thank you for your suggestion regarding the inclusion of additional tasks, such as StarCraft II or Hanabi. We want to clarify that our paper does not evaluate on only two tasks, but on two broad categories of tasks: **information exchange** and **multi-agent debate**. Within each category, we evaluate across multiple specific tasks. In total, we conduct experiments on **8 tasks**, with the MMLU benchmark even containing **57 sub-tasks**. We argue that this constitutes a **comprehensive and varied evaluation**. This number and diversity of tasks in our evaluation are already extensive compared to many other works in LLM-based MAS.
>
> While we agree that tasks like **StarCraft II** and **Hanabi** would be valuable to explore, these are relatively less studied in the context of LLM-based MAS. We argue that the benchmarks we’ve chosen are sufficient for demonstrating the effectiveness of Optima.
>
> We appreciate your suggestion and agree that these tasks could serve as interesting directions for future work, and we are currently considering them for follow-up experiments.
>
> ---
>
> **2. Number of Agents in Experiments**
>
> We understand your concern regarding the limited number of agents in our experiments. As the **first work to optimize multi-agent communication via iterative training**, we chose to focus on the cleanest scenario involving two agents. This allowed us to focus on the core challenges of optimizing communication efficiency without introducing the additional complexities that would arise with more agents, such as communication topology, speaking order, and coordination between multiple agents.
>
> Still, we recognize the importance of demonstrating the scalability of Optima to more agents. As such, we have conducted additional experiments:
>
> 1. We directly applied the model trained in the 2-agent scenario to a 3-agent communication setting:
>
> |  | **Hotpot QA** |  | **2WMH QA** |  |
> | --- | --- | --- | --- | --- |
> | **Setting** | **Perf** | **#Toks** | **Perf** | **Toks** |
> | CoT | 20.5 | 139.8 | 25.6 | 123.7 |
> | CoT-SC | 28.7 | 1052.8 | 33.8 | 996.3 |
> | MAD | 25.9 | 543.7 | 28.4 | 570.9 |
> | iSFT | 47.5 | 99.1 | 53.2 | 121.2 |
> | iDPO | 46.3 | **71.1** | 57.8 | 108.6 |
> | iSFT-DPO | **50.4** | 97.6 | **69.2** | **76.6** |
>
> Although the performance slightly decreased compared to the 2-agent setting, this is expected due to the increased complexity of information exchange in multi-agent scenarios, where more agents require more sophisticated coordination. This result demonstrates the **generalizability** of the trained models to scenarios with additional agents.
>
> 1. Since we assign one agent as solver, and the other as critic in multi-agent debate task, it is hard to generalize the already trained model to more agent scenarios. Therefore, we additionally train the models on both types of tasks in the most basic 3-agent scenario, where agents speak in order. We show that Optima still offers satisfying performance:
>
> |  | **2WMH QA** |  | **ARC-C** |  |
> | --- | --- | --- | --- | --- |
> |  | **Perf** | **#Toks** | **Perf** | **#Toks** |
> | CoT | 20.5 | 139.8 | 65.2 | 138.9 |
> | CoT-SC | 28.7 | 1052.8 | **75.6** | 1116.7 |
> | MAD (2-agent) | 25.9 | 543.7 | 71.4 | 478.0 |
> | iSFT | **62.0** | 62.8 | 72.6 | 123 |
> | iDPO | 56.3 | 55.8 | **75.6** | 76.2 |
> | iSFT-DPO | 60.7 | **53.7** | 75.4 | **72.7** |
>
> As anticipated, in the information exchange task, the 3-agent setting generally performs worse than the 2-agent setting due to the more distributed nature of the information, but Optima still offers performance gain against baselines. In the debate task, Optima also continues to provide a performance boost while significantly reducing token usage. As noted earlier, the 3-agent setting introduces additional complexities, such as speaking order and communication topology, which need to be carefully managed for optimal performance. We believe that with improved configuration and adjustments in the 3-agent setup, Optima can still be effectively utilized to train models in these scenarios.
>
> We take your point seriously, and we believe that these additional experiments will help demonstrate the adaptability and scalability of Optima to scenarios involving more agents.

---

> > ### Author Response · Authors · 2024-11-20
> >
> > **3. Performance on Other Base Models**
> >
> > Regarding your concern about the performance of Optima on other base models, we have already conducted additional experiments on the **Llama 3.2 3B model**. The results are as follow:
> >
> > |  | **HotpotQA** |  | **2WMH** |  | **MATH** |  | **GSM8k** |  | **ARC-C** |  |
> > | --- | --- | --- | --- | --- | --- | --- | --- | --- | --- | --- |
> > | **Setting** | **Perf** | **#Toks** | **Perf** | **#Toks** | **Perf** | **#Toks** | **Perf** | **#Toks** | **Perf** | **#Toks** |
> > | CoT | 22.7 | 355.8 | 16.5 | 235 | 0.463 | 556.7 | 78.7 | 288.9 | 51.5 | 256.1 |
> > | CoT-SC | 28.0 | 2804.6 | 24.2 | 467.7 | **0.568** | 4436 | **88.6** | 2300.4 | 57.6 | 2068.6 |
> > | MAD | 31.8 | 1677.9 | 27.6 | 2152.8 | 0.463 | 2509.2 | 81.2 | 763.8 | 37.4 | 872.4 |
> > | iSFT | **53.2** | 54 | 65.2 | **47.7** | 0.461 | 585.4 | **81.8** | 313.9 | 62.7 | 156.2 |
> > | iDPO  | 49.4 | 59.9 | 57.0 | 65.4 | **0.474** | 575.7 | 81.4 | 290.8 | **63.1** | **132.7** |
> > | iSFT-DPO | 52.5 | **48.7** | **66.8** | 51.4 | 0.468 | **548.4** | 80.8 | **270.1** | 61.6 | 141.4 |
> >
> > These results confirm that Optima consistently achieves superior performance with significant token efficiency, even on a different model. We will add the experiments and relevant analysis in the appendix, and mention it in the main text.
> >
> > ---
> >
> > We believe that Optima represents a significant step forward in optimizing communication efficiency in multi-agent systems, and we appreciate your constructive feedback. We are committed to addressing the concerns raised and will update the paper accordingly.
> >
> > We look forward to the possibility of continuing this discussion and improving the manuscript further. If our clarification addresses your concerns, we kindly ask you to consider raising the score. Thank you once again for your time and consideration.

---

> > > ### Author Response · Authors · 2024-11-25
> > >
> > > Dear Reviewer uBkL,
> > >
> > > We sincerely thank you for your constructive review and positive remarks regarding the clarity and quality of our work. In our rebuttal, we have provided additional experimental results and clarifications to address your concerns about task diversity, agent scalability, and performance on different base models.
> > >
> > > As the discussion phase is nearing its end, we kindly ask for your feedback on our responses. If you feel that we have adequately addressed your concerns, we would appreciate it if you could consider raising your score.
> > >
> > > In the general response, we have also emphasized the novelty and impact of our contributions, particularly in highlighting token efficiency as a critical issue in multi-agent systems and exploring inference scaling laws as an important research direction. We hope these contributions are reflected in your assessment of our work.
> > >
> > > Thank you once again for your time and support in improving our paper.
> > >
> > > Best regards,
> > >
> > > Authors

---

> > > > ### Author Response · Authors · 2024-11-27
> > > > **Request for Response**
> > > >
> > > > Dear Revewer uBkL,
> > > >
> > > > We deeply appreciate the time and effort you have invested in reviewing our work. In response to your feedback, we have conducted additional experiments and clarified key points. We would be grateful for any further suggestions or comments you might have.
> > > >
> > > > Best,
> > > >
> > > > Authors

---

> > > > > ### Author Response · Authors · 2024-12-03
> > > > > **Request for Feedback**
> > > > >
> > > > > Dear Reviewer uBkL,
> > > > >
> > > > > Since it is the last day of the discussion, we hope that you can take a look at our response. Thanks.
> > > > >
> > > > > Best regards,
> > > > >
> > > > > The Authors

---

### Official Review · Reviewer_YHqV · 2024-10-24

**Soundness:** 2
**Presentation:** 3
**Contribution:** 1
**Rating:** 5
**Confidence:** 4

**Summary:**

This paper investigates inter-agent communication and task inference effectiveness within LLM-based MAS. It introduces a framework based on an iterative generate, rank, select, and train paradigm to address these challenges. The core of this framework is iteration, and build iSFT and iDPO based on the iteration paradigm. The iSFT leverages prompt formats to create a diverse dataset. Subsequently, it removes these formats and fine-tunes the model based on the generated trajectories. The iDPO employs Monte Carlo Tree Search (MCTS) to generate diverse data through multi-agent conversations (MAD), alternating between MCTS-based data generation and model updates using DPO.
The authors evaluate this framework across several benchmarks using Llama 3 8B as the baseline model. They achieve promising results compared to Chain-of-Thought (CoT), SC (n=8), MAD, and AutoForm.

**Strengths:**

The method is written with significant detail, which is easy to follow.

Introducing an effective and efficient LLM-based frameework, which can be treated as a foundational model, seems important and interesting.

Experiments are provided for various environments and demonstrate promising results.

**Weaknesses:**

1. Unfair Comparison: The methods in OPTIMA do show less token consumption and higher accuracy compared to other approaches. However, the comparisons between iSFT, iDPO, or iSFT-DPO and other methods may be unfair. The authors did not clarify whether OPTIMA’s training is conducted online or offline (I assume it is offline). If it is online, the token consumption should not be lower than CoT (since MCTS requires searching across 24 trajectories). If it is offline, then the token consumption comparison is unfair, as the OPTIMA methods fine-tune the model on diverse prompts or multiple sampled data, which would have already consumed a significant amount of tokens during the post-training phase. The paper, however, only compares token consumption during the inference process.

2. Lack of Novelty: The methods iSFT and iDPO seem to lack innovation. iDPO merely combines MCTS from ToT with DPO, while iSFT simply adds a step of supervised fine-tuning (SFT) after removing the prompt. These methods seem incremental rather than novel, similar approaches can already be found, such as [1], [2], [3], [4], [5].

3. Reward Function: The authors mention the reward function multiple times, but it is only briefly defined in line 146. Meanwhile, in line 79, the reward is described as the core element of OPTIMA’s success. This undermines the credibility of the proposed method since the reward function is not thoroughly explained or explored.

4. Scalability: The authors frequently mention the scalability of OPTIMA, which seems exaggerated. There is no detailed discussion of the framework's ability to scale up in the experiments. Additionally, the token consumption during fine-tuning with iSFT and iDPO would increase significantly as the number of agents grows, which could limit the scalability of the framework.

[1]Wu T, Li X, Liu P. Progress or regress? self-improvement reversal in post-training[J]. arXiv preprint arXiv:2407.05013, 2024.

[2]Pang R Y, Yuan W, Cho K, et al. Iterative reasoning preference optimization[J]. arXiv preprint arXiv:2404.19733, 2024.

[3] Mukobi G, Chatain P, Fong S, et al. SuperHF: Supervised Iterative Learning from Human Feedback[J]. arXiv preprint arXiv:2310.16763, 2023.

[4]Xiong W, Dong H, Ye C, et al. Iterative preference learning from human feedback: Bridging theory and practice for rlhf under kl-constraint[C]//Forty-first International Conference on Machine Learning. 2024.

[5] Pang R Y, Yuan W, Cho K, et al. Iterative reasoning preference optimization[J]. arXiv preprint arXiv:2404.19733, 2024.

**Questions:**

As the weakness mentioned.

---

> ### Author Response · Authors · 2024-11-20
>
> We thank you for your thoughtful review of our work. We appreciate your positive remarks regarding the clarity and detail of our framework, as well as the promising results we presented. However, we respectfully disagree with some of the assessments, particularly regarding the novelty of our work, the fairness of our comparisons, and the scalability of our approach. We would like to address your concerns in detail.
>
> ---
>
> **1. Unfair Comparison: Token Consumption During Inference and Post-Training**
>
> We understand your concern about comparing token consumption during inference while ignoring the tokens consumed during post-training. However, our major goal in this work is to demonstrate the **efficiency of the final system during inference**, which is the key metric for real-world deployment. Since our objective is to optimize token efficiency during inference, we believe that **focusing on inference tokens is entirely reasonable**.
>
> While we agree that the training cost is important, **it is not relevant to our claim regarding inference efficiency**. To clarify, the post-training process consumes a certain amount of tokens, but this cost is negligible compared to pre-training tokens. More importantly, the improvements in inference efficiency outweigh the costs incurred during our Optima post-training. Specifically:
>
> - For **iSFT**, the training (including initialization and 5 iterations) can be completed in under **12 hours** using 8 A100 GPUs for almost all the tasks.
> - For **iDPO**, due to the MCTS component, the training takes longer, but can still be completed within **24 hours** on 8 A100 GPUs for almost all the tasks.
>
> These timeframes, given the substantial gains in inference efficiency, are highly acceptable and reflect a **reasonable tradeoff**. Requiring the inclusion of post-training tokens in the efficiency comparison is **unreasonable**, as it diverges from the usual practice in the field, where post-training costs are typically not included in such comparisons. This is especially true when the focus is on optimizing inference efficiency.
>
> Furthermore, if we were to include training tokens when comparing with baselines, then **no training-based method would appear effective**, as the number of tokens consumed during post-training is generally much larger than the tokens used during inference. If the evaluation is restricted to inference-only techniques without additional training tokens, such as prompt engineering, the scope for meaningful improvements in efficiency would be **highly constrained**. Such techniques, while lightweight, tend to provide limited improvements and would restrict the allowable methods to **superficial approaches** that fail to address the deeper challenges of optimizing efficiency.
>
> ---
>
> **2. Lack of Novelty**
>
> We respectfully disagree with the assessment that our work lacks novelty. While iSFT and iDPO have been explored in previous works, our contribution lies in the **novel problem focus**, the **novel findings**, and the **novel application** of these techniques. Specifically:
>
> - **Novel Problem Focus**: Our work addresses the critical but under-explored issue of communication efficiency in LLM-based MAS. This challenge, which has been long overlooked, is central to the successful deployment of multi-agent systems, especially when operating at scale.
> - **Novel Findings**: We demonstrate that our approach leads to systems that **improve inference scaling laws**. This finding extends beyond the original goals of iSFT and iDPO (focusing only on effectiveness), opening new research directions regarding the optimization of LLM-based MAS, and even more broadly, the LLM systems.
> - **Novel Application**: Our approach applies iSFT and iDPO to multi-agent systems, which requires adaptations to the reward design, data construction, and filtering processes. These modifications are crucial for ensuring that the techniques are effective in MAS scenarios, making our approach both innovative and impactful.
>
> We believe that these contributions, combined with our comprehensive empirical evaluation, position our work as a significant advancement in the field, rather than an incremental improvement.

---

> > ### Author Response · Authors · 2024-11-20
> >
> > **3. Brief Introduction to the Reward Function**
> >
> > We appreciate your feedback on the reward function and agree that further explanation is warranted. We will revise the manuscript to include a **more detailed explanation** of the reward function, including the following:
> >
> > - A clear formulation of the readability constraint $R_\text{loss}$ and the intuition behind it.
> > - The rationale for using this metric as a proxy for readability, which provides **high computational efficiency** while capturing the essential aspects of model communication quality.
> >
> > Additionally, we argue that our **ablation studies** (Section 3.4 and Appendix Table 5) has empirically demonstrate the importance of each term in the reward function, which has shown the essentiality of the reward designing.
> >
> > ---
> >
> > **4. Scalability**
> >
> > We believe there may be a misunderstanding regarding the term “scaling law” as used in our paper. Our discussion primarily focuses on **inference scaling laws**, which refer to the **relationship between inference compute and model performance**—a concept extensively studied in prior LLM research [1,2,3]. This is distinct from **scalability** in terms of the number of agents in a multi-agent system.
> >
> > To clarify, we do not claim to address the scalability of the number of agents in this work. Instead, our focus is on optimizing the **token consumption per sampled response**, thereby achieving a more favorable tradeoff between inference compute and performance. The concerns regarding the agent-number scalability of Optima do not directly apply to our claims or the scope of this paper.
> >
> > We hope this clarification resolves any confusion and provides a better understanding of our paper’s focus. Additionally, if you are interested in the agent-number scalability of Optima, we have included relevant experimental results in our responses to Reviewer Kkwh and Reviewer uBKL. These results demonstrate that models trained in a 2-agent scenario can be effectively transferred to a 3-agent setting, and that training models directly in 3-agent scenarios using Optima also yields satisfying performance.
> >
> > ---
> >
> > We hope that our responses clarify the concerns raised and allow you to reconsider the assessment of our work. We appreciate your careful review and your constructive feedback. Thank you once again for your time and consideration.
> >
> > [1] Chen, Lingjiao, et al. "Are more llm calls all you need? towards scaling laws of compound inference systems." *arXiv preprint arXiv:2403.02419* (2024).
> >
> > [2] Wu, Yangzhen, et al. "An empirical analysis of compute-optimal inference for problem-solving with language models." *arXiv preprint arXiv:2408.00724* (2024).
> >
> > [3] Brown, Bradley, et al. "Large language monkeys: Scaling inference compute with repeated sampling." *arXiv preprint arXiv:2407.21787* (2024)

---

> > > ### Author Response · Authors · 2024-11-25
> > >
> > > Dear Reviewer YHqV,
> > >
> > > Thank you for your detailed review and for raising important points regarding our work. We have carefully addressed all your concerns in our rebuttal, including providing new experimental results and additional explanations on the scalability of our framework and the novelty of our contributions.
> > >
> > > As there are only about two days left for discussion, we would greatly appreciate your feedback on our responses. If you believe we have satisfactorily addressed your concerns, we kindly ask you to consider raising your score.
> > >
> > > In the general response, we have also highlighted the novelty of addressing token efficiency in multi-agent systems and our pioneering exploration of inference scaling laws as a practical alternative to increasing training compute. We hope these contributions align with your expectations for impactful research.
> > >
> > > Thank you once again for your valuable comments and time.
> > >
> > > Best regards,
> > >
> > > Authors

---

> > > > ### Comment · Reviewer_YHqV · 2024-11-25
> > > > **Further response to authors#13477**
> > > >
> > > > We appreciate the authors' detailed reply. However, some concerns, particularly regarding **Question 1**, remain unresolved. These concerns are crucial as they affect the soundness of the work, primarily due to the lack of theoretical guarantees supporting the claims. This highlights the importance of ensuring fairness, accuracy, and comprehensiveness in the experimental evaluations.
> > > >
> > > > ## Regarding Question 1: Unfair Comparison - Token Consumption During Inference and Post-Training
> > > >
> > > > As the authors mentioned:
> > > >
> > > > > For iSFT, the training (including initialization and 5 iterations) can be completed in under 12 hours using 8 A100 GPUs for almost all the tasks.
> > > > > For iDPO, due to the MCTS component, the training takes longer, but can still be completed within 24 hours on 8 A100 GPUs for almost all the tasks.
> > > >
> > > > However, other approaches compared in the work remain **prompt-based**, such as **CoT MAD** and **self-consistency**, which rely on the vanilla base model to construct MAS. Under such circumstances, a more reasonable approach would involve fine-tuning the base model of these methods and conducting performance comparisons accordingly. Additionally, introducing **MCTS-based methods** for comparison would greatly enhance the credibility of the results.
> > > >
> > > > ---
> > > >
> > > > ## On Token Consumption
> > > >
> > > > We acknowledge the authors’ explanation that the primary objective of this work is to demonstrate the efficiency of the final system during inference, as inference efficiency is the key metric for real-world deployment. However, we believe that neglecting token consumption during post-training raises concerns about the fairness of the comparisons. The authors state:
> > > >
> > > > > While we agree that the training cost is important, it is not relevant to our claim regarding inference efficiency. To clarify, the post-training process consumes a certain amount of tokens, but this cost is negligible compared to pre-training tokens. More importantly, the improvements in inference efficiency outweigh the costs incurred during our Optima post-training.
> > > >
> > > > While this argument highlights the efficiency gains during inference, it does not address the broader issue of comparative fairness. Specifically:
> > > >
> > > > 1. Fine-tuning approaches, such as those applied to the baseline methods, would require including post-training token costs for a fair and direct comparison.
> > > > 2. The authors argue that post-training token costs are typically excluded in efficiency comparisons in the field. While this may be common practice, it is not a justification for bypassing a more rigorous and equitable evaluation in this specific case.
> > > >
> > > > To further clarify the point: if post-training tokens are excluded, the comparison may disproportionately favor methods like **iSFT** and **iDPO**, which incorporate training stages to optimize inference efficiency. Conversely, approaches that rely purely on inference-time techniques, such as prompt engineering, may be unfairly disadvantaged, even though they remain lightweight and practical in certain deployment scenarios.

---

> ### Author Response · Authors · 2024-11-26
>
> Thank you for your follow-up response and for acknowledging the importance of inference efficiency and the improvements achieved by Optima. We appreciate this ongoing discussion and would like to address your remaining concerns. Before proceeding, we believe it would be helpful to review and clarify the key points from both sides.
>
> ---
>
> ### 1. **Clarifying Our Points of Divergence**
>
> We believe we have reached consensus on the key achievements of Optima: it produces more effective and efficient models, with significant gains in inference efficiency. However, our perspectives diverge on whether training tokens should be included in efficiency comparisons:
>
> - **Our position:** Since the goal of Optima is to optimize inference efficiency, it is reasonable to compare Optima-trained models with plain models, focusing solely on inference metrics. This approach aligns with standard practices in the field, such as comparing RLHF-trained models to SFT models based on chat ability, without including training tokens in the comparison.
> - **Your concern:** Additional training introduces new costs, so training tokens should be included, or inference-time MCTS methods should be incorporated as baselines for fairness.
>
> ---
>
> ### 2. **Addressing the Inclusion of Training Tokens**
>
> We understand your concern about including training tokens, but we believe this perspective may overlook key principles of evaluating training-based methods:
>
> 1. Optima's goal is to create inference-efficient models. Just as RLHF comparisons focus on chat ability without accounting for training tokens, we evaluate inference efficiency because it reflects our method’s primary target. Including training tokens dilutes the focus on this core objective.
> 2. It is a well-established practice to exclude training tokens in comparisons. **This norm exists because post-training methods are primarily evaluated based on their ability to inject desired properties into the model. As long as the required resources are practical**—e.g., Optima training takes under 12–24 hours on 8 A100 GPUs—**these methods are considered effective**. We encourage you to reflect on why this practice is widely accepted and how it aligns with Optima’s focus.
> 3. As we have said, if training tokens are included, **no training-based method would appear efficient**. For example, training-based methods like RLHF would always seem inferior in token usage compared to lightweight inference-only techniques like prompt engineering.
>
> ---
>
> ### 3. **Addressing the Suggestion to Include MCTS-Based Methods**
>
> Including MCTS-based methods as inference-time baselines is inappropriate for several reasons:
>
> - MCTS is inherently inefficient for inference, as it relies on generating multiple trajectories for each response. This contradicts Optima’s goal of improving inference efficiency.
> - **MCTS requires access to the correct answer or a reliable reward signal to evaluate trajectories, which is unavailable during inference**. Therefore, incorporating MCTS-based methods would not align with the practical objectives of our work.
>
> ---
>
> ### 4. **Training Tokens in Context of Ablation Studies**
>
> Our ablation study can be seen as a comparison between Optima method and plain iSFT and iDPO, which also fine-tune the models. The findings in our ablation study show that removing token-efficiency-related reward terms from Optima leads to models with significantly higher token usage or inferior performance.
>
> Thus, our ablation studies provide evidence that Optima acts as an effective tradeoff between the effectiveness and efficiency when compared with other fine-tuning methods such as iSFT and iDPO.
>
> ---
>
> ### 5. **The Complementarity of Training and Inference Techniques**
>
> We want to emphasize that **training-based methods like Optima are not in conflict with inference-time techniques** such as self-consistency or MCTS. Instead, **they are complementary**:
>
> - As shown in **Table 1 and Figure 3** of our paper, models trained with Optima can still **benefit from inference-time techniques, further improving performance and efficiency**.
> - **Combining training-based and inference-time methods allows for better results than either approach alone**.
>
> **We respectfully disagree with the notion that these techniques should be viewed as mutually exclusive.**

---

> > ### Author Response · Authors · 2024-11-26
> >
> > ### 6. **Additional Experiment: Fine-Tuning Baselines**
> >
> > To further address your concerns about fine-tuning, we conducted experiments using GSM8k and MATH datasets, as only these two datasets provide human-labeled CoT-like solutions. We did grid search over the hyperparameters:
> >
> > - lr: [1e-6, 5e-6, 1e-5]
> > - epoch: 1, 2, 3
> > - batch: 128
> >
> > The best results are presented below:
> >
> > |  | **MATH** |  | **GSM8k** |  |
> > | --- | --- | --- | --- | --- |
> > | **Setting** | **Performance** | **#Tokens** | **Performance** | **#Tokens** |
> > | CoT (base) | 23.9 | 329.8 | 71.5 | 230.9 |
> > | CoT-SC (base) | 35.7 | 2600.9 | 80.3 | 1828.7 |
> > | CoT (trained) | 24.0 | 225.4 | 68.3 | 101.6 |
> > | CoT-SC (trained) | 32.0 | 1795.8 | 78.8 | 817.9 |
> >
> > Interestingly, fine-tuned models showed only marginal performance changes, with slight drops in some cases. This suggests that Llama has already been extensively trained on high-quality math data, limiting the impact of additional fine-tuning.
> >
> > It is important to note that this experiment is inherently **unfair to Optima**, as it leverages high-quality human-annotated solutions, while Optima relies **solely on self-improvement**. Despite this, Optima demonstrates efficiency and effectiveness gains.
> >
> > ---
> >
> > ### 7. **Revisiting Fairness in Comparisons**
> >
> > Lastly, **we respectfully disagree with the accusation that our comparisons are “unfair.”** Including training tokens would obscure the value of training-based methods, especially **given their potential to be combined with inference-time techniques**. As shown in our experiments, **Optima is complementary to these techniques**, providing a holistic approach to improving inference efficiency. **Instead of viewing these methods as opposing, we encourage considering how they can work together to advance the field.**
> >
> > ---
> >
> > We hope this comprehensive response addresses your concerns and clarifies our perspective. Thank you for engaging in this discussion and providing valuable feedback.

---

### Official Review · Reviewer_GSSx · 2024-11-04

**Soundness:** 3
**Presentation:** 3
**Contribution:** 2
**Rating:** 5
**Confidence:** 4

**Summary:**

This work proposes a framework that improve the effectiveness and efficiency of LLMs in multi-agent dialogue systems by iteratively optimizing data and training LLMs using SFT/DPO. For iSFT, it use sampling to generate better data; for iDPO, it use MCTS to generate paired data. Additionally, a carefully designed reward function ensures the system's overall effectiveness and efficiency in task performance.

**Strengths:**

The paper is well-structured and easy to understand, addressing an important problem with a clearly articulated methodology. The data generation mechanism for MAS is interesting and may facilitate further improvements in MAS, including frameworks like AutoGen.

**Weaknesses:**

1.	While this paper offers a well-structured approach to enhancing the effectiveness and efficiency of MAS, iSFT and iDPO have been extensively explored in prior works. Thus, the main contributions here—reward function design and data improvement mechanism in MAS—offer limited novelty within the existing research landscape.

2.	A thorough and fair experimental comparison is very important if the novelty is limited. Another key weakness of this paper is unfair comparison. As this framework has trained LLM to maximize a delicate reward, it’s slightly unfair to compare with prompt-tuning method like CoT, SC, etc.

3.	This paper focuses on LLM-based MAS, but the definition of the problem and the specific methods do not adequately reflect the multi-agent aspect. It appears that the multi-agent scenario is only evaluated in the experiments. The methods are relatively general and do not address the key issues in MAS.

Other trivial weakness in this paper:

1.	Fact error in abstract: iSFT and iDPO with reward filters can be considered as RL, but SFT and DPO are not RL.

2.	The language modeling loss in equation 1 is not defined.

**Questions:**

1.	Figure 3(a) need further elaboration. How does Optima influence MAS’s inference scaling law comparing to baseline?
2.	I would appreciate further discussion on the advantage of optimizing the effectiveness and efficiency of a MAS versus optimizing these aspects of a single LLM using similar reward functions.
3.	Would improvements in effectiveness and efficiency facilitate the design of communication topologies within MAS?

---

> ### Author Response · Authors · 2024-11-20
>
> Thank you for your thorough review and constructive feedback. We appreciate your recognition of our paper's clear structure and methodology. We would like to address your concerns and clarify several important points:
>
> ---
>
> 1. **Novelty and Contribution**
>
> We respectfully disagree with the assessment of limited novelty. While iSFT and iDPO have indeed been explored previously, our work's novelty lies in:
>
> - **Novel Problem Focus**: We identify and address the critical but long-ignored problem of communication efficiency in LLM-based MAS. To our knowledge, we are the first to systematically study and optimize this aspect.
> - **Novel Findings**: We demonstrate that our approach yields systems with better inference scaling laws, a finding that extends **beyond the original goals** of these training algorithms and **opens new research directions**.
> - **Novel Application**: The application of these algorithms to multi-agent scenarios represents an innovation. The challenges of maintaining effective communication while optimizing for efficiency require adaptation of these techniques, such as reward designing, reward balancing, data construction process and data filtering.
>
> The significance of our work lies not just in the methods used, but in identifying and successfully addressing a crucial challenge in MAS deployment.
>
> ---
>
> 2. **Experimental Comparisons**
>
> We appreciate your concern about comparison fairness. Our comparisons aim to demonstrate that Optima can successfully **inject the ability to be both concise and accurate into models.** To achieve this, we compared trained and untrained models to show that the framework performs as intended. However, we take your point seriously, and conducted additional experiments by training the models on the MATH and GSM8k training set to evaluate the impact of training on the training data, as they have human annotated CoT-like data. Note that this comparison may be unfair to us, as we only use model-generated answer to do self-improve, while training directly on the annotated CoT data benefit from human annotation. We do grid search over the hyperparameters:
>
> - lr: [1e-6, 5e-6, 1e-5]
> - epoch: 1, 2, 3
> - batch: 128
>
> We found the results with lr=1e-6, epoch=2, bsz=128 for MATH, and lr=1e-6, epoch=3, bsz=128 for GSM8k are the best:
>
> |                | MATH |        | GSM8K | |
> | --- | --- | --- | --- | --- |
> | **Baseline** | **Perf** | **#Toks** | **Perf** | **#Toks** |
> | CoT (base) | 23.9 | 329.8 |  71.5 | 230.9 |
> | CoT-SC (base) | 35.7 | 2600.9 | 80.3 | 1828.7 |
> | CoT (trained) | 24.0 | 225.4 | 68.3 | 101.6 |
> | CoT-SC (trained) | 32.0 | 1795.8 | 78.8 | 817.9 |
>
> Interestingly, we did not observe a significant difference between the performance of trained and untrained models. This is possibly because Meta has already extensively trained Llama on high-quality math data. This suggests that the observed improvements may not be solely attributable to training on the training set. We appreciate your feedback, as it helps us further clarify the robustness of Optima's contributions.
>
> ---
>
> 3. **Lack of Multi-Agent Aspects**
>
> Thank you for raising this point. Still, we believe our work is fundamentally grounded in multi-agent scenarios:
>
> - Our framework specifically targets multi-agent communication efficiency and effectiveness
> - Both evaluation settings (information exchange and multi-agent debate) represent typical multi-agent scenarios
> - The reward function and training procedure are designed specifically for multi-agent interaction
>
> While our methods have potential applications beyond MAS, we view this generalizability as a strength rather than a limitation.
>
> ---
>
> 4. **Technical Writing Corrections**
>
> We appreciate your careful attention to detail. We will revise the abstract to avoid mischaracterizing SFT and DPO as RL methods, and add explicit formula for the language modeling loss.

---

> > ### Author Response · Authors · 2024-11-20
> >
> > 5. **Figure 3(a) Clarification**
> >
> > Figure 3(a) demonstrates that models trained with Optima maintain desirable inference scaling properties, consistent with previous findings about performance improving with increased inference compute. The gap between coverage (ideal performance upper bound) and majority voting suggests significant room for further improvements.
> >
> > Figure 3(b) specifically shows Optima's impact on inference scaling laws compared to baselines, demonstrating comparable performance with only ~20% of the inference compute.
> >
> > ---
> >
> > 6. **Advantages of Optimizing Multi-Agent vs Single-Agent**
> >
> > Many scenarios where we apply our optimization framework are inherently multi-agent in nature. For instance, the information exchange tasks involved in our paper, and other tasks such as embodied agent collaboration, are fundamentally constrained by information asymmetry or physical limitations that naturally require multiple agents. In such cases, optimization can only meaningfully occur in the multi-agent context. So compare MAS with single agent is inappropriate.
> >
> > For reasoning tasks that we explore in the multi-agent debate setting, while these could be performed by a single LLM, previous work has demonstrated advantages of multi-agent approaches [1,2,3]. Given this established benefit, we chose to focus our optimization efforts on the multi-agent setting to potentially amplify these advantages. While we acknowledge that similar optimization techniques might benefit single LLM scenarios, our primary focus on multi-agent systems led us to prioritize that direction in this work. We view the potential generalizability of our method to single-agent scenarios as an additional strength rather than a limitation, though we leave this exploration for future work.
> >
> > ---
> >
> > 7. **Would improving effectiveness and efficiency facilitates the design of communication topologies?**
> >
> > We would say thinking the point in reverse could be more interesting - how to optimize communication topologies for enhanced effectiveness and efficiency. While our current work focuses on optimizing the content of communication to facilitate better MAS, we believe researchers could tackle this challenge from the perspective of communication topology, which would be an exciting direction to explore.
> >
> > We think our work opens up many promising research directions in MAS, and by highlighting the importance of efficiency in these systems, we make them more practical for real-world applications. Given these broader implications and contributions, we kindly ask you to reconsider your assessment of our work. We believe it represents a valuable addition to the field that could inspire and enable future research. Thank you again for your careful review.
> >
> > ---
> >
> > [1] Chan, Chi-Min, et al. "Chateval: Towards better llm-based evaluators through multi-agent debate." *arXiv preprint arXiv:2308.07201* (2023).
> >
> > [2] Wu, Qingyun, et al. "Autogen: Enabling next-gen llm applications via multi-agent conversation framework." *arXiv preprint arXiv:2308.08155* (2023).
> >
> > [3] Du, Yilun, et al. "Improving factuality and reasoning in language models through multiagent debate." *arXiv preprint arXiv:2305.14325* (2023).

---

> > > ### Author Response · Authors · 2024-11-25
> > >
> > > Dear Reviewer GSSx,
> > >
> > > We deeply appreciate your detailed review and insightful comments. In our rebuttal, we have worked to address your concerns comprehensively, including clarifications on fairness in comparisons, the novelty of our contributions, and the multi-agent aspects of our framework.
> > >
> > > We kindly ask if you could provide your feedback on our responses, especially since there are only about two days remaining in the discussion phase. Additionally, we have emphasized the novelty of our work in the general response, particularly the focus on communication efficiency in LLM-based MAS and the exploration of inference scaling laws as a practical and impactful research direction. We hope these contributions align with your assessment of our work.
> > >
> > > Thank you again for your time and constructive input.
> > >
> > > Best regards,
> > >
> > > Authors

---

> > > > ### Author Response · Authors · 2024-11-27
> > > > **Request for Response**
> > > >
> > > > Dear Reviewer GSSx,
> > > >
> > > > We have carefully addressed your concerns, added new experiments, and clarified specific details in our response. We value your insights and look forward to your comments.
> > > >
> > > > Best regards,
> > > >
> > > > Authors

---

> > > > > ### Comment · Reviewer_GSSx · 2024-11-30
> > > > >
> > > > > I appreciate the authors' effort in providing additional experiments. The results are compelling and effectively demonstrate the proposed reward's efficacy. I agree that addressing the communication efficiency problem is important, and as such, I have decided to increase my score to 5.

---

> > > > > > ### Author Response · Authors · 2024-11-30
> > > > > >
> > > > > > We are very grateful for your acknowledgment of the compelling results and the efficacy of the proposed reward mechanism, as well as the importance of the addressed problem. Thank you for raising the score!
> > > > > >
> > > > > > Still, we would like to kindly remind you that a score of 5 still means "weak reject". If you do feel that our work addresses an important question and presents compelling results, please allow us to kindly ask you to consider raising your score further to 6 (marginally above the acceptance threshold) or higher. We truly believe our work is solid and important enough to be accepted. Such a score would not only reflect your recognition of the importance of the problem but also provide support for research that aims to advance the inference efficiency of multi-agent systems and more broadly, LLM systems.
> > > > > >
> > > > > > If there are any additional aspects you believe should be addressed to make this work suitable for acceptance, please let us know. Thank you once again for your careful evaluation and constructive feedback, which have been invaluable in improving our work. We truly appreciate your consideration and support!

---

### Official Review · Reviewer_Kkwh · 2024-11-09

**Soundness:** 3
**Presentation:** 3
**Contribution:** 4
**Rating:** 6
**Confidence:** 4

**Summary:**

The paper presents a collaborative training method for systems of LLMs, Optima. Optima features a carefully crafted reward function that incentivizes the communications readibility, task performance, and token efficiency. Additionally, a variant of Optima employs MCTS to sample diverse trajectories. Trajectories are then trained on using SFT or DPO. The authors evaluate their algorithm on diverse complex reasoning and information-asymmetric question answering tasks and observe performance improvements in both tasks performance and token efficiency.

**Strengths:**

The authors tackle an extremely timely problem with significant community interest - how to optimise LLM agents for collaborative tasks.
There are several algorithmic innovations, and the empirical evaluation is extensive.
The observed improvements are significant.
Last but not least, there is a very extensive treatment of related work.

**Weaknesses:**

* lack of theoretical examination of why the reward function used is working
* lack of evaluation of models other than llama 8bn (other/smaller models would have been interesting to see).

**Questions:**

* what about systemic biases/safety misalignment arising from training?
* can we extend this to systems of more than 2 agents, and how does this scale?
* could you provide assurance that results are not from training on test data (lacks details)

---

> ### Author Response · Authors · 2024-11-20
>
> Thank you for your thoughtful review and for recognizing the timeliness and significance of the challenge our work addresses. We also appreciate your positive assessment of our extensive empirical evaluation. We are pleased to address your specific concerns:
>
> ---
>
> 1. **Theoretical Explanation of Reward Function**
>
> Our reward function design follows the empirical approach common in traditional RL literature (e.g., +1 or +10 for successful trajectories and -1 or 0 for failures). While primarily empirically-driven, our design still follows clear principled considerations:
>
> - The token efficiency term ($R_\text{token}$) directly incentivizes concise communication, while the language modeling loss term ($R_\text{loss}$) promotes natural and interpretable exchanges. High rewards indicate that a model achieves accuracy (via $R_\text{task}$), conciseness (via $R_\text{token}$), and readability (via $R_\text{loss}$), which is desired for our final goal.
> - Our ablation studies (Table 3) demonstrate each component's crucial role - removing either term significantly degrades either efficiency or performance.
>
> **This empirically-validated approach aligns with successful practices in RL applications where reward shaping is often guided by desired behaviors rather than theoretical guarantees.**
>
> ---
>
> 2. **Results on Other Models**
>
> We have conducted additional experiments using **Llama 3.2 3B on several benchmark** to address your concern about the performance of Optima on different models. The results, as shown below, demonstrate Optima’s consistent advantages in both effectiveness and efficiency compared to the baselines:
>
> |  | **HotpotQA** |  | **2WMH** |  | **MATH** |  | **GSM8k** |  | **ARC-C** |  |
> | --- | --- | --- | --- | --- | --- | --- | --- | --- | --- | --- |
> | **Setting** | **Perf** | **#Toks** | **Perf** | **#Toks** | **Perf** | **#Toks** | **Perf** | **#Toks** | **Perf** | **#Toks** |
> | CoT | 22.7 | 355.8 | 16.5 | 235 | 0.463 | 556.7 | 78.7 | 288.9 | 51.5 | 256.1 |
> | CoT-SC | 28.0 | 2804.6 | 24.2 | 467.7 | **0.568** | 4436 | **88.6** | 2300.4 | 57.6 | 2068.6 |
> | MAD | 31.8 | 1677.9 | 27.6 | 2152.8 | 0.463 | 2509.2 | 81.2 | 763.8 | 37.4 | 872.4 |
> | iSFT | **53.2** | 54 | 65.2 | **47.7** | 0.461 | 585.4 | **81.8** | 313.9 | 62.7 | 156.2 |
> | iDPO  | 49.4 | 59.9 | 57.0 | 65.4 | **0.474** | 575.7 | 81.4 | 290.8 | **63.1** | **132.7** |
> | iSFT-DPO | 52.5 | **48.7** | **66.8** | 51.4 | 0.468 | **548.4** | 80.8 | **270.1** | 61.6 | 141.4 |
>
> These results confirm that Optima consistently achieves superior performance with significant token efficiency, even on a different model. We will add the experiments and relevant analysis in the appendix, and mention it in the main text.
>
> ---
>
> 3. **Safety and Alignment Considerations**
>
> While safety alignment isn't the primary focus of Optima (which targets task effectiveness and efficiency), we take your concern seriously. We follow similar approach proposed in LLM-as-judge [1] and conduct additional analysis using GPT-4o, asking it to judge whether the communication trajectory contains unsafe content. Given the high cost of evaluating all trajectories, we follow the principles of sample size determination for proportions to estimate the proportion of unsafe content with 99% confidence. Specifically, for each dataset, we sample $n\times \sqrt{\frac{N-n}{N-1}}$ samples, where $n=\frac{2.576^2\cdot 0.5^2}{(1-0.99)^2}$ and $N$ is the dataset size. This formula ensures that the calculated error rate on the sampled trajectories falls within the 99% confidence interval of the error rate for the entire dataset. After conducting this analysis, we found **no unsafe content** in any of the sampled communication trajectories.
>
> We will add the analysis and the prompt to our appendix to improve our work. Thank you for your advice.

---

> ### Author Response · Authors · 2024-11-20
>
> 4. **Scalability**
>
> To assess the scalability of Optima, we directly transferred the model trained under the 2-agent scenario to a 3-agent setting. The results are summarized below:
>
> |  | **Hotpot QA** |  | **2WMH QA** |  |
> | --- | --- | --- | --- | --- |
> | **Setting** | **Perf** | **#Toks** | **Perf** | **Toks** |
> | CoT | 20.5 | 139.8 | 25.6 | 123.7 |
> | CoT-SC | 28.7 | 1052.8 | 33.8 | 996.3 |
> | MAD | 25.9 | 543.7 | 28.4 | 570.9 |
> | iSFT | 47.5 | 99.1 | 53.2 | 121.2 |
> | iDPO | 46.3 | **71.1** | 57.8 | 108.6 |
> | iSFT-DPO | **50.4** | 97.6 | **69.2** | **76.6** |
>
> Although the performance slightly decreased compared to the 2-agent setting, this is expected due to the increased complexity of information exchange in multi-agent scenarios, where more agents require more sophisticated coordination. This result demonstrates the **generalizability** of the trained models to scenarios with additional agents.
>
> Since we assign one agent as solver, and the other as critic in multi-agent debate task, it is hard to generalize the already trained model to more agent scenarios. Therefore, we additionally train the models on both types of tasks in the most basic 3-agent scenario, where agents speak in order. We show that Optima still offers satisfying performance:
>
> |  | **2WMH QA** |  | **ARC-C** |  |
> | --- | --- | --- | --- | --- |
> |  | **Perf** | **#Toks** | **Perf** | **#Toks** |
> | CoT | 20.5 | 139.8 | 65.2 | 138.9 |
> | CoT-SC | 28.7 | 1052.8 | **75.6** | 1116.7 |
> | MAD (2-agent) | 25.9 | 543.7 | 71.4 | 478.0 |
> | iSFT | **62.0** | 62.8 | 72.6 | 123 |
> | iDPO | 56.3 | 55.8 | **75.6** | 76.2 |
> | iSFT-DPO | 60.7 | **53.7** | 75.4 | **72.7** |
>
> As anticipated, in the information exchange task, the 3-agent setting generally performs worse than the 2-agent setting due to the more distributed nature of the information, but Optima still offers performance gain against baselines. In the debate task, Optima also continues to provide a performance boost while significantly reducing token usage. As noted earlier, the 3-agent setting introduces additional complexities, such as speaking order and communication topology, which need to be carefully managed for optimal performance. We believe that with improved configuration and adjustments in the 3-agent setup, Optima can still be effectively utilized to train models in these scenarios.
>
> ---
>
> 5. **Data Integrity Verification**
>
> To address concerns about potential test data leakage:
>
> - We strictly adhered to **official train-test splits** for all datasets
> - We additionally conduct verification by checking for any overlap between training and test questions, and we find **no instances of test data appearing in training sets**
>
> This confirms the validity of our reported performance improvements. We will add it to the appendix. We appreciate your attention to this important methodological consideration.
>
> ---
>
> **Our Results' Broader Impact**
>
> Beyond addressing specific concerns, we'd like to again argue that Optima represents an important advancement in LLM-based MAS and LLM systems more broadly:
>
> - We achieve up to 90% reduction in token usage while maintaining or improving performance, opening new possibilities for **practical deployment**. We are the first to highlight the importance of **simultaneously** optimizing effectiveness and efficiency, and propose an effective training framework for this purpose.
> - The improved inference scaling laws suggest **better compute utilization**, making multi-agent systems more **viable** for real-world applications.
> - Our successful application of iterated self-improvement opens promising new research directions about training-based self-improvement of LLM MAS.
>
> We believe these contributions, combined with our thorough empirical validation and careful attention to methodological concerns, make Optima a valuable addition to the field.
>
> ---
>
> We welcome any requests for clarification or additional questions. If our responses have adequately addressed your concerns, we kindly ask you to consider raising the score. We truly appreciate your careful review.
>
> ---
>
> [1] Zheng, Lianmin, et al. "Judging llm-as-a-judge with mt-bench and chatbot arena." Advances in Neural Information Processing Systems 36 (2023): 46595-46623.

---

> > ### Author Response · Authors · 2024-11-25
> >
> > Dear Reviewer Kkwh,
> >
> > We have carefully addressed all the points you raised in our rebuttal, including additional experiments and clarifications, such as results on different models and the scalability of Optima.
> >
> > We would greatly appreciate it if you could take a moment to provide feedback on our responses, as there are only about two days left in the discussion phase. Additionally, we kindly ask you to consider raising your score if you feel that we have adequately addressed your concerns.
> >
> > We have also highlighted the novelty and significance of our contribution in the general response, particularly regarding token efficiency in multi-agent systems and the importance of inference scaling laws. We hope these points resonate with you as key contributions to the field.
> >
> > Thank you again for your time and for helping us improve our work.
> >
> > Best regards
> >
> > Authors

---

> > > ### Author Response · Authors · 2024-11-27
> > >
> > > Dear Reviewer Kkwh,
> > >
> > > We sincerely invite you to review our response. We have made every effort to address your concerns by conducting additional experiments and providing further clarifications on key points. We would greatly appreciate your feedback.
> > >
> > > Best regards,
> > >
> > > The Authors

---

> > > > ### Author Response · Authors · 2024-12-03
> > > > **Request for Feedback**
> > > >
> > > > Dear Reviewer Kkwh,
> > > >
> > > > Since it is the last day of the discussion, we hope that you can take a look at our response. Thanks.
> > > >
> > > > Best regards,
> > > >
> > > > The Authors

---

### Author Response · Authors · 2024-11-20
**General Responses to All Reviewers**

We sincerely thank all the reviewers for their thoughtful feedback and for recognizing various aspects of our work. We would like to take this opportunity to emphasize the novelty and significance of the problem we address, which we feel underpins the contributions of our paper.

---

**Highlighting the Novelty of the Addressed Problem**

Our work is among the **first to highlight the importance of token efficiency** in multi-agent systems (MAS). Communication efficiency is a critical, yet underexplored, aspect of MAS, and addressing it is key to making these systems more practical for real-world applications. Furthermore, we are the **first to leverage training-based methods** to jointly optimize token efficiency and task performance in this context, setting our approach apart from inference-only techniques, such as prompt engineering, which often offer limited improvements.

---

**Importance of Inference Scaling Laws**

Inference scaling laws, which analyze the relationship between inference compute and model performance, are garnering increasing attention in the research community. Unlike adding training compute, which requires substantial resources and time, increasing inference compute has been shown to be a **highly effective alternative** for improving LLM systems. This direction is gaining recognition as an important pathway for achieving better performance without the prohibitive costs of additional training.

Our work makes a significant contribution by demonstrating how reducing inference token usage positively impacts **inference-time scaling laws**, leading to improved efficiency and performance. We believe that the extensive and solid empirical results presented in our paper clearly reveal the benefits of this approach.

---

**Feasibility and Impact of Our Contribution**

Beyond addressing the theoretical significance, we also provide a **feasible and easy-to-understand pipeline** to achieve these benefits, making our methods accessible to the broader research community. By presenting a practical framework for reducing inference tokens while maintaining or improving task performance, our work opens new possibilities for optimizing MAS and paves the way for future research in this critical area.

---

We hope that reviewers will consider the novelty and importance of the problem we address, along with the strong empirical evidence supporting our contributions. We believe that our work represents a meaningful advancement in the field and contributes to the growing interest in optimizing inference efficiency and performance in LLM-based MAS.

Thank you again for your careful review and for engaging with our work. We welcome any additional questions or suggestions and look forward to further improving our manuscript based on your valuable feedback.

---

### Author Response · Authors · 2024-11-25
**Request for Feedback**

Dear reviewers,

We sincerely thank you for your thoughtful comments and valuable feedback on our submission. We have carefully addressed all the concerns and questions raised, providing detailed responses and additional experimental results where necessary. Your insights have significantly helped us refine and strengthen our work.

As the discussion phase is nearing its conclusion, we kindly request your feedback on our responses at your earliest convenience. Your input is invaluable in ensuring that we address any remaining concerns effectively. If there are any aspects of our responses that require further clarification or additional experiments, we would be happy to provide them within the remaining time.

Thank you again for your time and effort in reviewing our paper. We greatly appreciate your engagement and look forward to your thoughts.

Best,

Authors

---

### Meta-Review · Area_Chair_goSU · 2024-12-20

**Metareview:**

This paper proposes a training method for multi-agent systems of LLMs. Unfortunately reviewers were concerned with novelty and incrementality of the approach.

**Additional Comments On Reviewer Discussion:**

This appears to be a very borderline paper: 6, 5, 5, 6, and several things jumped out at me as I read through the discussion. First, one of the higher scoring reviewers made the kind of mistake in their review that would lead me to discount it (they suggested inappropriate baselines in a way that made me think they are not familiar with the area of research, and note that they also checked the box to indicate they are reviewing for the first time). Second, there was no real discussion here. Most of the reviewers did not engage, and did not even provide any indication that they read the author responses. So my reasoning to that we should not accept it is based on two things: (1) discounting the reviewer who appeared not to be familiar with the area, and (2) the similarity of the issues raised by the two reviewers who gave lower scores.

---

### Decision · Program_Chairs · 2025-01-22

Reject